# Asymptotically Best Causal Effect Identification with Multi-Armed Bandits

**Alan Malek**
DeepMind London
alanmalek@deepmind.com

**Silvia Chiappa**
DeepMind London
csilvia@deepmind.com

## Abstract

This paper considers the problem of selecting a formula for identifying a causal quantity of interest among a set of available formulas. We assume an sequential setting in which the investigator may alter the data collection mechanism in a data-dependent way with the aim of identifying the formula with lowest asymptotic variance in as few samples as possible. We formalize this setting by using the best-arm-identification bandit framework where the standard goal of learning the arm with the lowest loss is replaced with the goal of learning the arm that will produce the best estimate. We introduce new tools for constructing finite-sample confidence bounds on estimates of the asymptotic variance that account for the estimation of potentially complex nuisance functions, and adapt the best-arm-identification algorithms of LUCB and Successive Elimination to use these bounds. We validate our method by providing upper bounds on the sample complexity and an empirical study on artificially generated data.

## 1 Introduction

Many scientific disciplines, including biology, healthcare, and social and behavioral sciences, are concerned with estimating the causal effect of some exposure on an outcome of interest through an observational study. When the structure of the causal relationships between relevant variables is known and satisfies certain conditions, the investigator can use identification formulas derived from the do-calculus to express a causal quantity $\tau$ as a functional of the distribution of the observations [23, 24], after which an estimate of $\tau$ can be obtained with an appropriate estimator[1]. For many structures, $\tau$ can be identified using several formulas involving different variables. In most real-world applications, the costs of measuring different variables can vary considerably. For instance, different medical tests can be more or less expensive or difficult to perform. Ideally, the investigator should conduct the observational study using the formula that optimally balances statistical performance with observational cost.

Recently introduced graphical criteria allow the comparison of identification formulas derived from the popular adjustment criterion w.r.t. the asymptotic variance of linear and nonparametric estimators [7, 27, 29, 31]. However, information about the causal graph structure alone does not allow comparison of certain adjustment formulas nor of arbitrary identification formulas. In addition, such criteria only focus on statistical performance and disregard observational cost. A selection method that is applicable to arbitrary identification formulas and accounts for both statistical performance and observational cost must make use of data.

A naïve approach to choose a formula would be to collect a dataset in which each observation includes all variables needed by all formulas, and then compare the cost-adjusted statistical performances of

---

[1]For simplicity of exposition, we assume that an identification formula is associated with only one estimator, but the method proposed in this paper can also be used in the more general case of several estimators.

35th Conference on Neural Information Processing Systems (NeurIPS 2021).

the formulas. We instead propose a sequential strategy in which the investigator decides, observation-by-observation, which subset of variables to observe with the goal of identifying the best formula with the fewest observations. We cast this strategy into the best-arm-identification bandit framework [20], by considering each formula as one arm and by replacing the typical goal of learning the arm with the best mean with the goal of learning the formula with the best long-run cost-adjusted statistical performance. This enables us to introduce methods for implementing the strategy by leveraging algorithms from the bandit literature.

To enable us to meaningfully compare long-run behavior of different estimators, we focus on $\sqrt{n}$-consistent estimators, i.e. with error $o_p(1/\sqrt{n})$, and use the asymptotic variance, which is the leading constant in the error rate [22], as the performance evaluation metric. We adapt well-known bandit algorithms to this goal by introducing finite-sample confidence intervals on the asymptotic variance. In particular, we show how to obtain such intervals for asymptotically linear estimators with known influence functions, which form a large class of $\sqrt{n}$-consistent estimators that can be constructed for any causal effect [15]. Estimators of causal quantities typically contain high-dimensional nuisance functions $\eta$, and modern practice is to estimate such functions using large model classes that do not have classical Donsker-like smoothness (e.g. neural networks), which often leads to a slow convergence rate of $\mathcal{O}(n^{-1/4})$ and therefore to loosing asymptotic linearity. Recent work in double machine learning has established Neyman orthogonality as a sufficient condition for asymptotic linearity even when $\eta$ is estimated at rate $\mathcal{O}(n^{-1/4})$ [2]. This result allows us to include estimators with complex nuisance functions in our setting. A critical aspect of our setting is that the asymptotic variance needs to be estimated at the same rate as $\tau$; otherwise the samples needed to estimate the asymptotic variance could be larger than the samples needed to directly estimate $\tau$ with an arbitrary estimator. Our main theorems show that, even without Neyman orthogonality, $\mathcal{O}(n^{-1/2})$ estimation of the asymptotic variance is possible whenever $\mathcal{O}(n^{-1/2})$ estimation of $\tau$ is possible.

The rest of the paper is organized as follows. Section 2 introduces our technical assumptions on the estimators and formalizes selection of the best formula as a best-arm-identification problem in multi-armed bandits. Section 3 shows how to construct finite-sample confidence sequences (i.e. confidence intervals that hold uniformly over sequences of random variables) on the asymptotic variance, which are then used in Section 4 to adapt the LUCB and Successive Elimination bandit algorithms to our setting. Sample complexity upper bounds are also derived. Finally, Section 5 presents an empirical evaluation of our methods on artificially generated data, showing significant sample complexity reduction with respect to a naïve uniform sampling method.

## 2 Causal Effect Identification with Multi-Armed Bandits

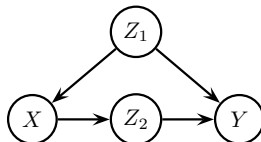

Let $X, Y$, and $\mathcal{V}$ be random variables with $X$ denoting an exposure, $Y$ an outcome of interest, and $\mathcal{V}$ a set of observable covariates. Let $p$ and $p_{\mathrm{do}(x)}$ indicate the *observational* and *interventional* distributions, respectively, which differ in passively observing $X$ versus intervening on $X$ by fixing its value to $x$.

We consider the *causal contrast* $\tau := \sum_x \lambda_x \mu_{\mathrm{do}(x)}$, where $\mu_{\mathrm{do}(x)} := \mathbb{E}_{p_{\mathrm{do}(x)}}[Y|X = x]$ and $\lambda_x$ is a scalar, in the overidentified setting in which $\mu_{\mathrm{do}(x)}$ can be expressed as a functional of the observational distribution with $K$ *identification formulas*, each using a different subset of covariates $\mathcal{Z}_k \subseteq \mathcal{V}$. For example, in the causal DAG above (see Appendix A), $\mu_{\mathrm{do}(x)}$ can be identified with the *adjustment criterion* using $Z_1$, i.e. as $\mu_{\mathrm{do}(x)} = \mathbb{E}_p[\mu_x(Z_1)]$ with $\mu_x(Z_1) := \mathbb{E}_p[Y|X = x, Z_1]$, or with the *frontdoor criterion* using $Z_2$, i.e. as $\mu_{\mathrm{do}(x)} = \sum_{z_2} p(Z_2 = z_2|X = x) \sum_{x'} p(X = x') \mu_{x'}(Z_2 = z_2)$.

We assume that each identification formula is associated with only one estimator $\hat{\tau}_k$ of $\tau$ mapping a dataset $\{w_k^i\}_{i=1}^n$ of samples from $p(W_k := (X, Y, \mathcal{Z}_k))$, each having *cost* $c_k$, to an estimate of $\tau$.

The goal of this paper is to propose a method for identify the estimator $\hat{\tau}_{k^*}$ with the best cost-adjusted statistical performance, using a sequential strategy in which the investigator chooses an estimator $\hat{\tau}_k$, collects an observation of $W_k$, and uses it to update its belief about the estimators. The investigator continues this process until the optimal estimator can be identified with high confidence. The aims is to concentrate observations on the most promising estimators and to identify the best estimator in as few observations as possible.

**Assumptions on Estimators of $\tau$.** We focus on estimators that are $\sqrt{n}$-*consistent*, i.e. with error $o_p(1/\sqrt{n})$, and compare their statistical performance by using the *asymptotic variance*, which is the leading constant in the error rate [22]. To make this comparison, our proposed algorithms in Section 4 require finite-sample confidence intervals on the asymptotic variance. While any finite-sample confidence interval can be employed, we show how to construct such intervals for a very common and often-studied class of $\sqrt{n}$-consistent estimators, namely *asymptotically linear estimators* with known *influence functions* in the presence of *nuisance functions*.

Nuisance functions are quantities that are required for the estimation of $\tau$, but of no interest otherwise. For example, the *augmented inverse probability weighted (AIPW) estimator* using covariates $\mathcal{Z} \subseteq \mathcal{V}$, defined as $\hat{\tau}(\mathcal{D}) = \mathbb{E}_{\mathcal{D}}\left[\sum_x \lambda_x \left(\frac{I_x(X)}{e_x(\mathcal{Z})}(Y - \mu_x(\mathcal{Z})) + \mu_x(\mathcal{Z})\right)\right]$, where $I_x$ is the indicator function, $e_x(\mathcal{Z}) := p(X = x|\mathcal{Z})$, and $\mathbb{E}_{\mathcal{D}}[\cdot]$ denotes empirical expectation w.r.t. dataset $\mathcal{D}$ of samples from $p(W := (X, Y, \mathcal{Z}))$, has nuisance function $\eta = (\mu_x(\mathcal{Z}), e_x(\mathcal{Z}))$.

**Definition 1.** *Let $\mathcal{D} = \{w^i\}_{i=1}^n$ be a dataset of samples from $p(W)$. An estimator $\hat{\tau}$ of $\tau$ with nuisance function $\eta$ is asymptotically linear [11, 30] if there exists a function $\phi$, called influence function (IF), satisfying $\mathbb{E}_p[\phi(W, \eta, \tau)] = 0$ and $\mathbb{E}_p[\phi^2(W, \eta, \tau)] < \infty$, and such that $\sqrt{n}(\hat{\tau}(\mathcal{D}) - \tau) = \frac{1}{\sqrt{n}}\sum_{i=1}^n \phi(w^i, \eta, \tau) + o_p(1)$. By the central limit theorem, $\hat{\tau}$ is $\sqrt{n}$-consistent and asymptotically normal with asymptotic variance $\sigma^2 = \mathbb{E}_p[\phi^2(W, \eta, \tau)]$.*

*If the influence function can be decomposed as $\phi(W, \eta, \tau) = \psi(W, \eta) - \tau$, then $\psi$ is called the uncentered influence function (UIF). In this case, $\tau = \mathbb{E}_p[\psi(W, \eta)]$ and $\sigma^2 = \mathrm{var}_p[\psi(W, \eta)]$.*

The important property of asymptotically linear estimators is that they have a well-understood asymptotic variance, and the UIF case provides additional structure.

When we need to estimate $\eta$, asymptotic linearity of $\hat{\tau}$ might not hold depending on the function class used to model the estimator $\hat{\eta}$ and on the rate at which $\hat{\eta}$ converges to $\eta$: a rate slower than $\mathcal{O}(n^{-1/2})$ could cause the same rate for the error of $\hat{\tau}$ and, therefore, a loss of asymptotically linearity. For example, if $\hat{\eta}$ is modelled with a high-dimensional or nonparametric function class, one generally expects rate $\mathcal{O}(n^{-1/4})$. Recent work in double machine learning [2] has shown that Neyman orthogonality is a sufficient condition for asymptotic linearity of $\hat{\tau}$ even when $\hat{\eta}$ converges at rate $\mathcal{O}(n^{-1/4})$. Jung et al. [15, 16] show that asymptotically linear estimators with Neyman orthogonality exist for all identification formulas. Hence, all identification formulas can be included in our framework even with nuisance functions estimators that converge at rate $\mathcal{O}(n^{-1/4})$.

**Multi-Armed Bandit Formalism.** We formalize the problem of identifying the estimator with the best cost-adjusted asymptotic variance as a *best-arm-identification* (BAI) in multi-armed bandits (MAB) problem. MAB is a powerful framework for modeling sequential decision problems under uncertainty as a repeated game between an investigator and the environment (see Appendix B).

For $k \in [K] := \{1, \ldots, K\}$, let $\hat{\tau}_k$ be an asymptotically linear estimator of $\tau$ with asymptotic variance $\sigma_k^2$ and cost $c_k$. The goal of the investigator is to identify the estimator $\tau_{k*}$ such that $k^* := \arg\min_k c_k \sigma_k^2$. This scaling arises because guaranteeing $|\hat{\tau}_k - \tau| = \mathcal{O}(\epsilon)$ with high probability requires $n = \mathcal{O}(\sigma_k^2/\epsilon^2)$ samples, which has a cost of $\mathcal{O}(c_k \sigma_k^2/\epsilon^2)$. For some $\delta > 0$, and $\epsilon > 0$, the goal is to find an $(\epsilon, \delta)$-PAC index $\hat{k}$, i.e. one that satisfies

$$\mathbb{P}\left(c_{\hat{k}}\sigma_{\hat{k}}^2 \geq \min_k c_k \sigma_k^2 + \epsilon\right) \leq \delta.$$

In words, $\hat{k}$ has probability at least $1 - \delta$ of being at most $\epsilon$-suboptimal.

At each round $n = 1, 2, \ldots$, the investigator chooses an index $k_n \in [K]$, obtains an observation $w_{k_n}^n \sim p(W_k)$, updates its belief about the cost-adjusted asymptotic variance $c_k \sigma_k^2$, and decides whether to continue sampling or return an $(\epsilon, \delta)$-PAC index.

The standard BAI goal is to identify the arm with the smallest average loss in as few samples as possible. Our setting can be treated as a variant by replacing this goal with the goal of learning the estimator with the lowest cost-adjusted asymptotic variance. This enables us to leverage algorithms from the BAI literature.

Most BAI algorithms fall into the following three categories: (1) action elimination algorithms, such as *Successive Elimination* (SE) [4], which keep a set of arms that could be optimal and alternates

between sampling all the arms in this set and using confidence intervals to prune the set; (2) *optimistic algorithms*, including Upper Confidence Bound (UCB) and *LUCB* [17], which first construct the most optimistic problem instance that is consistent with the confidence intervals (e.g. by assuming the smallest mean values) and then act greedily as if the instance was true; (3) Track-and-Stop-style algorithms [19], which compute an asymptotic lower bound on the sample complexity and try to keep the empirical sampling proportion close to the sampling proportion achieving the lower bound.

Track-and-Stop-style algorithms require the sampling distribution of all the arms to be from a parametric family and therefore are not appropriate for our setting. Instead, action elimination and optimistic algorithms can be implemented whenever one has finite-sample confidence sets. We focus on LUCB and SE, as both algorithms can be implemented using arbitrary confidence sets (i.e. that are not a known function of $n$).

In the next section, we develop the necessary tools for estimating two-sided confidence bounds on $\sigma_k^2$ that hold for all estimators and all rounds simultaneously.

# 3 Finite-Sample Confidence Sequences for the Asymptotic Variance

We now turn towards constructing a finite-sample confidence sequence on the asymptotic variance $\sigma^2$ of a particular estimator $\hat{\tau}$ of $\tau$ that depends on assumptions on the distribution of $W$, smoothness properties of the influence function $\phi$, and the rate of convergence of $\hat{\eta}$.

## 3.1 Confidence Sequences

A confidence sequence, defined for a stochastic process, essentially provides a confidence interval that holds uniformly over time.

**Definition 2.** *For a stochastic process $\{\xi_n\}_{n \geq 1}$, $\xi_n \in \mathbb{R}$, and coverage level $\alpha > 0$, a confidence sequence of level $\alpha$ is a sequence of real numbers $u := \{u_n\}_{n \geq 1}$, referred to as a boundary sequence at level $\alpha$, satisfying $\mathbb{P}(\forall n \geq 1 : \xi_n \leq u_n) > 1 - \alpha$.*

Instead of using a confidence sequence, one could control $\xi_n$ by defining a confidence interval for every $n$ and taking a union bound. However, to maintain a total probability of error of $\alpha$, the confidence interval for $\xi_n$ must have error probabilities that sum to at most $\alpha$, which typically results in a confidence interval width that is a logarithmic factor larger [13].

## 3.2 Sample-Splitting Estimator of $\sigma^2$

Let $\hat{\tau}$ be an estimator of $\tau$ with IF $\phi$, asymptotic variance $\sigma^2$, and nuisance function $\eta$ with estimator $\hat{\eta}$. We propose and analyze the following sample-splitting estimator $\hat{\sigma}^2$ of $\sigma^2$ on dataset $\mathcal{D}$. We first divide $\mathcal{D}$ into the three folds $\mathcal{D}^\eta$, $\mathcal{D}^\tau$ and $\mathcal{D}^\sigma$. We then use $\mathcal{D}^\eta$ to obtain an estimate $\hat{\eta}(\mathcal{D}^\eta)$ of $\eta$, compute $\hat{\tau}(\mathcal{D}^\tau, \hat{\eta}(\mathcal{D}^\eta))$ as an arbitrary solution in $\{\tau' : \mathbb{E}_{\mathcal{D}^\tau}[\phi(W, \hat{\eta}(\mathcal{D}^\eta), \tau')] = 0\}$, and evaluate

$$\hat{\sigma}^2(\mathcal{D}) := \mathbb{E}_{\mathcal{D}^\sigma}\left[\phi^2\left(W, \hat{\eta}(\mathcal{D}^\eta), \hat{\tau}(\mathcal{D}^\tau, \hat{\eta}(\mathcal{D}^\eta))\right)\right]. \tag{1}$$

In the case in which $\hat{\tau}$ has UIF $\psi$, the estimation procedure can be simplified by dividing $\mathcal{D}$ into two folds, $\mathcal{D}^\eta$ and $\mathcal{D}^\sigma$, and computing

$$\hat{\sigma}^2(\mathcal{D}) := \text{var}_{\mathcal{D}^\sigma}\left[\psi(W, \hat{\eta}(\mathcal{D}^\eta))\right] = \mathbb{E}_{\mathcal{D}^\sigma}\left[\phi^2\left(W, \hat{\eta}(\mathcal{D}^\eta), \hat{\tau}(\mathcal{D}^\sigma, \hat{\eta}(\mathcal{D}^\eta))\right)\right], \tag{2}$$

where $\text{var}_{\mathcal{D}^\sigma}$ indicates empirical variance w.r.t. $\mathcal{D}^\sigma$. In both cases, data splitting alleviates the bias in $\hat{\sigma}^2$ by forcing it to be independent of the bias in $\hat{\eta}$ (and of the bias of $\hat{\tau}$ in the IF case). The UIF case is more sample efficient because we do not need to directly compute $\hat{\tau}$.

## 3.3 Confidence Sequences for $|\hat{\sigma}^2(\mathcal{D}) - \sigma^2|$

We want to study the behavior of an estimator $\hat{\tau}$ evaluated on growing datasets, as one would find in a bandit problem. We consider datasets $\mathcal{D}_1 \subseteq \mathcal{D}_2 \subseteq \ldots$, where $\mathcal{D}_n$ is obtained by augmenting $\mathcal{D}_{n-1}$ with new samples. We also assume that the data folds grow in the same manner. For example, in the UIF case, we have $\mathcal{D}_n = \mathcal{D}_n^\eta \cup \mathcal{D}_n^\sigma$ with $\mathcal{D}_{n-1}^\eta \subseteq \mathcal{D}_n^\eta$, and $\mathcal{D}_{n-1}^\sigma \subseteq \mathcal{D}_n^\sigma$. We wish to obtain a confidence sequence on the estimation error of the asymptotic variance, $|\hat{\sigma}^2(\mathcal{D}_n) - \sigma^2|$, in terms the estimation error of the nuisance function $\|\hat{\eta}(\mathcal{D}_n^\eta) - \eta\|$ and other problem parameters.

The main result of this section is that the estimation error of the sample-splitting estimators defined in Eqs. (1) and (2) only scales with $\mathcal{O}(\|\hat{\eta}(\mathcal{D}_n^\eta) - \eta\|^2)$; this scaling allows for $|\hat{\sigma}^2(\mathcal{D}_n) - \sigma^2| = \mathcal{O}(n^{-1/2})$ even when $\|\hat{\eta}(\mathcal{D}_n^\eta) - \eta\| = \mathcal{O}(n^{-1/4})$. We present the UIF case first.

**Theorem 1.** *Consider an $L$-Lipschitz UIF $\psi$, an upper bound $\tilde{\tau}$ on $|\tau|$, and let $\alpha > 0$. Let $\{\mathcal{D}_n\}_{n \geq 1}$ be a sequence of datasets with $\mathcal{D}_n = \mathcal{D}_n^\eta \cup \mathcal{D}_n^\sigma$, $\mathcal{D}_{n-1}^\eta \subseteq \mathcal{D}_n^\eta$, and $\mathcal{D}_{n-1}^\sigma \subseteq \mathcal{D}_n^\sigma$. Assume that $u^{(\psi,1)}$, $u^{(\psi,2)}$, and $u^\eta$ are boundary sequences of level $\alpha$ for $\left|\mathbb{E}_{\mathcal{D}_n^\sigma}[\psi(W, \eta)] - \mathbb{E}[\psi(W, \eta)]\right|$, $\left|\mathbb{E}_{\mathcal{D}_n^\sigma}[\psi(W, \eta)^2] - \mathbb{E}[\psi(W, \eta)^2]\right|$, and $\|\hat{\eta}(\mathcal{D}_n^\eta) - \eta\|$, respectively. Then, the estimator $\hat{\sigma}^2$ defined in Eq. (2) satisfies*

$$\mathbb{P}\left(\forall n \geq 1 : \left|\hat{\sigma}^2(\mathcal{D}_n) - \sigma^2\right| \leq 2L^2(u_n^\eta)^2 + u_n^{(\psi,2)} + \left(u_n^{(\psi,1)}\right)^2 + 2\tilde{\tau}u_n^{(\psi,1)}\right) \geq 1 - 3\alpha.$$

A confidence sequence for $\|\hat{\eta}(\mathcal{D}_n^\eta) - \eta\|$ and tail control of $\psi(W, \eta)$ are sufficient for control of $\left|\hat{\sigma}^2(\mathcal{D}_n) - \sigma^2\right|$. Note that we only require control of $\psi(W, \eta)$ at the true $\eta$. Also note that $(u_n^\eta)^2$, rather than $u_n^\eta$, appears in the bound, which justifies our claim of $\mathcal{O}(\|\hat{\eta}(\mathcal{D}_n^\eta) - \eta\|^2)$ scaling. Computing this bound is an essential subroutine, outlined on the right.

---

**CSUpdate** (UIF version)

---

**Input** Boundary sequences $u := (u^\eta, u^1, u^2)$,
$\hat{\eta}, L, \tilde{\tau}, \mathcal{D}^\eta, \mathcal{D}^\sigma, \psi$
$\hat{\sigma}^2 \leftarrow \mathrm{var}_{\mathcal{D}^\sigma}[\psi(W, \hat{\eta}(\mathcal{D}^\eta))]$
Using $n_1 = |\mathcal{D}^\eta|$ and $n_2 = |\mathcal{D}^\sigma|$,
$\beta \leftarrow 2L^2(u_{n_1}^\eta)^2 + u_{n_2}^2 + (u_{n_2}^1)^2 + 2\tilde{\tau}u_{n_2}^1$
Return $\hat{\sigma}^2, \beta$

---

*Proof Outline.* Let $\hat{\eta}_n := \hat{\eta}(\mathcal{D}_n^\eta)$, $\hat{\sigma}_n^2 := \hat{\sigma}^2(\mathcal{D}_n)$, and $\mathbb{E}_n := \mathbb{E}_{\mathcal{D}_n^\sigma}$. Using the identity $\mathrm{var}_n[\psi(W, \hat{\eta}_n)] = \mathbb{E}_n[\psi(W, \hat{\eta}_n)^2] - \mathbb{E}_n[\psi(W, \hat{\eta}_n)]^2$, we can expand $\hat{\sigma}_n^2 - \sigma^2$, as

$$\hat{\sigma}_n^2 - \sigma^2 = \mathbb{E}_n\left[(\psi(W, \hat{\eta}_n) - \psi(W, \eta))^2\right] + 2\mathbb{E}_n\left[\psi(W, \eta)(\psi(W, \hat{\eta}_n) - \psi(W, \eta))\right]$$
$$- \mathbb{E}_n[\psi(W, \hat{\eta}_n) - \psi(W, \eta)]\mathbb{E}_n[\psi(W, \hat{\eta}_n) + \psi(W, \eta)]$$
$$+ \left(\mathbb{E}_n[\psi^2(W, \eta)] - \mathbb{E}[\psi^2(W, \eta)]\right) + \left(\mathbb{E}_n[\psi(W, \eta)]^2 - \mathbb{E}[\psi(W, \eta)]^2\right).$$

Since $\psi$ is $L$-Lipschitz, the first term can be bounded by $L^2 \|\hat{\eta}_n - \eta\|^2$. The second and third terms can be simplified with Cauchy-Schwarz, the first order terms cancel out, and the result is another $L^2 \|\hat{\eta}_n - \eta\|^2$ term. The forth term is controlled by $u_n^{(\psi,2)}$, and the final term can bounded as

$$\left|\mathbb{E}_n[\psi(W, \eta)]^2 - \mathbb{E}[\psi(W, \eta)]^2\right| \leq \left|\left(\mathbb{E}[\psi(W, \eta)] - u_n^{(\psi,1)}\right)^2 - \mathbb{E}[\psi(W, \eta)]^2\right|$$
$$\leq \left(u_n^{(\psi,1)}\right)^2 + 2\left|u_n^{(\psi,1)}\mathbb{E}[\psi(W, \eta)]\right| \leq \left(u_n^{(\psi,1)}\right)^2 + 2\tilde{\tau}u_n^{(\psi,1)}.$$

$\square$

A similar result can be obtained without an UIF as long as one has an additional confidence sequence on $\hat{\tau}(\mathcal{D}_n^\tau, \hat{\eta}(\mathcal{D}_n^\eta))$. See Appendix F for a precise statement and all proofs. In this case, the CSUpdate subroutine needs to be modified to use a third data fold $\mathcal{D}_n^\tau$, the appropriate $\hat{\sigma}^2$, and the confidence sequence from Theorem 2.

**Theorem 2.** *Consider an $L$-Lipschitz IF $\phi$, an upper bound $\tilde{\tau}$ on $|\tau|$, and let $\alpha > 0$. Let $\{\mathcal{D}_n\}_{n \geq 1}$ be a sequence of datasets with $\mathcal{D}_n = \mathcal{D}_n^\eta \cup \mathcal{D}_n^\tau \cup \mathcal{D}_n^\sigma$, $\mathcal{D}_{n-1}^\eta \subseteq \mathcal{D}_n^\eta$, $\mathcal{D}_{n-1}^\tau \subseteq \mathcal{D}_n^\tau$, and $\mathcal{D}_{n-1}^\sigma \subseteq \mathcal{D}_n^\sigma$. Assume that there exists boundary sequences of level $\alpha$ as in Theorem 1 with $\phi$ replacing $\psi$ throughout, and that $u^\tau$ is a boundary sequences of level $\alpha$ for $|\hat{\tau}(\mathcal{D}_n^\tau, \hat{\eta}(\mathcal{D}_n^\eta)) - \tau|$. Then, the estimator $\hat{\sigma}^2$ defined in Eq. (1) satisfies*

$$\mathbb{P}\left(\forall n \geq 1 : \left|\hat{\sigma}^2(\mathcal{D}_n) - \sigma^2\right| \leq 2L^2(u_n^\eta + u_n^\tau)^2 + u_n^{(\phi,2)} + \left(u_n^{(\phi,1)}\right)^2 + 2\tilde{\tau}u_n^{(\phi,1)}\right) \geq 1 - 4\alpha.$$

Combined with a result that $|\hat{\tau}(\mathcal{D}_n^\tau, \hat{\eta}(\mathcal{D}_n^\eta)) - \tau| = \mathcal{O}(\|\hat{\eta}(\mathcal{D}_n^\eta) - \eta\|^2)$ (see e.g. [2, 5]), we have that $|\hat{\sigma}^2(\mathcal{D}_n) - \sigma^2| = \mathcal{O}(\|\hat{\eta}(\mathcal{D}_n^\eta) - \eta\|^2)$.

We emphasize that the $\mathcal{O}(\|\hat{\eta}(\mathcal{D}_n^\eta) - \eta\|^2)$ scaling is surprising, as a naïve argument would produce a dependence on $\mathcal{O}(\|\hat{\eta}(\mathcal{D}_n^\eta) - \eta\|)$ instead; using the shorthand $\hat{\eta}_n = \eta(\mathcal{D}_n^\eta)$ and $\hat{\tau}_n = \hat{\tau}(\mathcal{D}_n^\tau, \hat{\eta}_n)$,

$$\hat{\sigma}^2(\mathcal{D}_n) - \sigma^2 = \mathbb{E}_n[\phi^2(W, \hat{\eta}_n, \hat{\tau}_n)] - \mathbb{E}[\phi^2(W, \eta, \tau)]$$
$$= \mathbb{E}_n\left[(\phi(W, \hat{\eta}_n, \hat{\tau}_n) - \phi(W, \eta, \tau))^2 + 2\phi(W, \eta, \tau)(\phi(W, \hat{\eta}_n, \hat{\tau}_n) - \phi(W, \eta, \tau))\right]$$
$$+ \mathbb{E}_n[\phi^2(W, \eta, \tau)] - \mathbb{E}[\phi^2(W, \eta, \tau)],$$

and the second term is $\mathcal{O}(\|\hat{\eta}_n - \eta\|)$ under a simple bound (e.g. Cauchy-Schwarz).

### 3.4 Sub-Gaussian Random Variables

Confidence sequences for empirical expectations around their means are well established in the literature, see e.g. [10]. As an illustration, this section derives a confidence sequence under the common assumption that the variables are sub-Gaussian.

**Definition 3.** *A random variable $W$ is $\lambda$ sub-Gaussian if there exists a constant $\lambda$ such that $\mathbb{E}\left[e^{t(W-\mathbb{E}[W])}\right] \leq e^{\lambda \frac{t^2}{2}}$ for all $t \in \mathbb{R}$. A random variable $W$ is $\nu$ sub-exponential with scale $c$ if there exist constants $\nu, c$ such that $\mathbb{E}\left[e^{t(W-\mathbb{E}[W])}\right] \leq e^{\nu \frac{t^2}{2}} \ \forall t \in [0, 1/c)$.*

A useful property is that the square of a sub-Gaussian random variable is sub-exponential. The specific parameters can be found by comparing moments; for example, [8, Appendix B] shows the following result.

**Lemma 1.** *If $W$ is $\lambda$ sub-Gaussian, then $W^2$ is $4\sqrt{2}\lambda^2$ sub-exponential with scale $c = 4\lambda$.*

Intuitively, sub-Gaussian random variables have tails no heavier than a Gaussian with variance $\lambda$, and sub-exponential random variables have tails no heavier than a $\chi^2$ distribution. Sub-Gaussianity is a common assumption in the bandit literature satisfied in many applications; for example, a $B$-bounded random variable is $B^2$ sub-Gaussian.

Confidence sequences for sub-Gaussian and and sub-exponential random variables can be found in the literature, see e.g. [10]. Then, we may use the fact that $\left|\mathbb{E}_{\mathcal{D}_n^\sigma}[\psi(W,\eta)] - \mathbb{E}[\psi(W,\eta)]\right|$ is sub-Gaussian and $\left|\mathbb{E}_{\mathcal{D}_n^\sigma}[\psi(W,\eta)^2] - \mathbb{E}[\psi(W,\eta)^2]\right|$ is sub-exponential to derive the following bound for our variance estimation error. See Appendix C for more details.

**Corollary 1.** *Let $\alpha \in (0,1)$ and assume the same setting as Theorem 1, and additionally that $\psi(W, \eta)$ is $\lambda$ sub-Gaussian. Then, for $\lambda' = \lambda \vee 8\lambda^2$, any $m > 0$, and $n' = (91\lambda'(\log(\lambda' n/m) + \log(1/\alpha))) \vee (m/\lambda')$, we can show that*

$$\mathbb{P}\left(\exists n \geq n' : \left|\hat{\sigma}^2(\mathcal{D}_n) - \sigma^2\right| \geq 2L^2(u_n^\eta)^2 + (3 + 6\tilde{\tau})\sqrt{\frac{\lambda'}{n}\left(\frac{1}{2}\log\left(\frac{\lambda' n}{m}\right) + \log\frac{2}{\alpha}\right)}\right) \leq \alpha.$$

### 3.5 Confidence Sequences for $\|\hat{\eta}(\mathcal{D}_n^\eta) - \eta\|$

Our main theorem presents a confidence sequence on the asymptotic variance in terms of the nuisance function estimation error, $\|\hat{\eta}(\mathcal{D}_n^\eta) - \eta\|$. There are many estimators $\hat{\eta}$ that have confidence sequences established in the literature, such as least squares estimators [3]. Additionally, for estimators with a confidence interval but no confidence sequence, we can always take a union bound over $n$.

To be precise, suppose that we have a function $R(n, \alpha)$ such that, for any $n \geq 1$ and $\alpha < 0$, $\mathbb{P}(\|\hat{\eta}(\mathcal{D}_n^\eta) - \eta\| \geq R(n, \alpha)) \leq \alpha$. Then $u_n^\eta = R(n, 6\alpha/(\pi n)^2)$ is a boundary sequence of level $\alpha$, verified by the calculation $\mathbb{P}(\exists n > 1 : \|\hat{\eta}(\mathcal{D}_n^\eta) - \eta\| \geq u_n^\eta) \leq \sum_{n=1}^\infty \mathbb{P}\left(\|\hat{\eta}(\mathcal{D}_n^\eta) - \eta\| \geq R(n, 6\alpha/(\pi n)^2)\right) \leq \sum_{n=1}^\infty \frac{6\alpha}{\pi n^2} \leq \alpha$.

Hence, without loss of generality, our algorithms and results can be stated in terms of a boundary sequence $u^\eta$. Typically, if $R(n, \alpha) = \mathcal{O}(n^\nu \log(1/\alpha))$, the above union bound construction only adds log terms to the final bound, which is generally sufficient for most applications.

## 4 CS-LUCB and CS-SE Algorithms

This section introduces our adaptations of the LUCB and SE bandit algorithms that use the confidence sequences derived in Section 3: we refer to them as CS-LUCB and CS-SE. For simplicity of exposition, we present the UIF case (the IF case requires keeping a third data fold and modifying CSUpdate as described in Section 3.3).

CS-LUCB and CS-SE are summarized in Algorithms 1 and 2. These algorithms take as input, for $k \in [K]$, the UIF $\psi_k$, the estimator $\hat{\eta}_k$ of the nuisance function $\eta_k$, the cost $c_k$, the boundary sequences $u_k := (u_k^\eta, u_k^1, u_k^2)$ and constants required by Theorem 1, and a batch size $B$.

At every round, CS-LUCB samples observations for the estimator with the lowest cost-adjusted asymptotic variance and for the estimator with the lowest lower confidence bound among the remaining estimators. Intuitively, samples from these estimators are the most informative. The algorithm stops when the confidence sequence of the best estimator is separated from the rest, up to the error tolerance.

CS-SE keeps a set $S$ of plausibly best estimators. At every round, it obtains samples for every estimator in $S$, updates their confidence sequences, and removes all estimators with lower bounds higher than the upper bound of the best from $S$, as these estimators are no longer plausibly optimal. The algorithm terminates when only one estimator remains or all remaining estimators are within $\epsilon$ of each other.

We define the gap of estimator $k$ to be $\Delta_k := c_k \sigma_k^2 - \min_{k'} c_{k'} \sigma_{k'}^2$. The random variable $\beta_k(n)$ is the confidence width returned by CSUpdate for estimator $k$ after $n$ updates, which might be random. We assume that $\beta_k(n)$ is independent of $\beta_j(n)$ for all $j \neq k$. In the remainder of the section, we prove that both algorithms return $(\epsilon, \delta)$-PAC indices if the confidence sequence width approaches zero and provide upper bounds on the sample complexities when an upper bound on the confidence width is available.

**Theorem 3.** *Assume that the conditions of Theorem 1 hold, that $u_{k,n}^\eta, u_{k,n}^{(\psi,1)}, u_{k,n}^{(\psi,2)} \to 0$ for all $k \in [K]$, and that all boundary sequences are of level $\alpha = \delta/(3K)$. Then both CS-LUCB and CS-SE with $u_k = (u_k^\eta, u_k^{(\psi,1)}, u_k^{(\psi,2)})$ return an $(\epsilon, \delta)$-PAC index.*

---

**Algorithm 1** CS-LUCB

**Input** $\epsilon > 0$, $B \geq 1$, $\tilde{\tau} > 0$, $L > 0$,
$\{\psi_k, \hat{\eta}_k, u_k, c_k : k \in [K]\}$
**for** *k=1,..., K* **do**
   Obtain $B$ new samples $\mathcal{D}$
   Add half of $\mathcal{D}$ to $\mathcal{D}_k^\eta$ and half to $\mathcal{D}_k^\sigma$
   $\hat{\sigma}_k^2, \beta_k \leftarrow \text{CSUpdate}(u_k, \hat{\eta}_k, \psi_k, L, \tilde{\tau}, \mathcal{D}_k^\eta, \mathcal{D}_k^\sigma)$
**end**
**for** $t = 1, 2, \ldots$ **do**
   $l_t \leftarrow \arg\min_{k \in [K]} c_k \hat{\sigma}_k^2$
   $u_t \leftarrow \arg\min_{k \neq l_t} c_k (\hat{\sigma}_k^2 - \beta_k)$
   **if** $c_{l_t}(\hat{\sigma}_{l_t}^2 + \beta_{l_t}) \leq c_{u_t}(\hat{\sigma}_{u_t}^2 - \beta_{u_t}) - \epsilon$ **then**
     Return $\hat{k} = l_t$
   **end**
   **for** $k \in u_t, l_t$ **do**
     Obtain $B$ new samples $\mathcal{D}$
     Add half of $\mathcal{D}$ to $\mathcal{D}_k^\eta$ and half to $\mathcal{D}_k^\sigma$
     $\hat{\sigma}_k^2, \beta_k \leftarrow \text{CSUpdate}(u_k, \hat{\eta}_k, \psi_k, L, \tilde{\tau}, \mathcal{D}_k^\eta, \mathcal{D}_k^\sigma)$
   **end**
**end**

---

**Algorithm 2** CS-SE

**Input** $\epsilon > 0$, $B \geq 1$, $\tilde{\tau} > 0$, $L > 0$,
$\{\psi_k, \hat{\eta}_k, u_k, c_k : k \in [K]\}$
$S \leftarrow [K]$, and $\mathcal{D}_k^\eta \leftarrow \emptyset, \mathcal{D}_k^\sigma \leftarrow \emptyset \; \forall k \in [K]$
**while** $|S| > 1$ **do**
   **for** $k \in S$ **do**
     Obtain $B$ new samples $\mathcal{D}$
     Add half of $\mathcal{D}$ to $\mathcal{D}_k^\eta$ and half to $\mathcal{D}_k^\sigma$
     $\hat{\sigma}_k^2, \beta_k \leftarrow \text{CSUpdate}(u_k, \hat{\eta}_k, \psi_k, L, \tilde{\tau}, \mathcal{D}_k^\eta, \mathcal{D}_k^\sigma)$
   **end**
   $l \leftarrow \arg\min_{k \in S} c_k \hat{\sigma}_k^2$
   $R \leftarrow \{k \in S : c_l(\hat{\sigma}_l^2 + \beta_l) \leq c_k(\hat{\sigma}_k^2 - \beta_k)\}$
   $S \leftarrow S \setminus R$
   **if** $\beta_k \leq \frac{\epsilon}{2}$ for all $k \in S$ **then**
     $S \leftarrow \arg\min_{k \in S} c_k \hat{\sigma}_k^2$
   **end**
**end**
Return $\hat{k} = S$

---

*If we have a deterministic upper bound $B_k(n, \delta)$ such that, for all $\delta > 0$, $\mathbb{P}(\beta_k(n) \leq B_k(n, \delta)) \geq 1 - \delta$, then both algorithms terminate in at most $\sum_{k \in [K]} \min\{n : B_k(n, \delta/K) \leq \frac{\Delta_k}{4} \vee \frac{\epsilon}{2}\}$ samples.*

*If, additionally, there exists constants $\nu_\eta$, $\nu_{(\psi,1)}$, and $\nu_{(\psi,2)}$ such that $u_{k,n}^\theta \leq \mathcal{O}(n^{-\nu_\theta} \log(nK/\delta))$ for all $\theta \in \{\eta, (\psi, 1), (\psi, 2)\}$ and all $k \in [K]$, then the sample complexity is*

$$\mathcal{O}\left(\sum_{k=1}^K (\Delta_k \vee \epsilon)^{-1/\nu} \left(\log \frac{K}{\delta(\Delta_k \vee \epsilon)^{1/\nu}}\right)^{1/\nu}\right),$$

*with probability at least $1 - \delta$, where $\nu = \min\{2\nu_\eta, \nu_{(\psi,1)}, \nu_{(\psi,2)}\}$. In particular, if $\psi(W, \eta)$ is sub-Gaussian, we recover the sample complexity results (up to log factors) of [4, 17] under the mild condition of $\nu_\eta \geq 1/4$.*

**Discussion.** To aid comparison with the BAI literature, we have provided sample complexity bounds for the special case in which $u_{k,n}^\theta = \mathcal{O}(n^{-\nu_\theta} \log(n/\delta))$. The most common assumption in the bandit literature is that the data is sub-Gaussian, meaning that we may take $\nu_\theta = \frac{1}{2}$. In this case, our sample

complexity results are within a log-factor of the optimal and within a log-factor of the lower bound when $\epsilon = 0$. Algorithms achieving the optimal rate include lil'UCB [14] and Exponential Gap Elimination [18]. Unfortunately, these algorithms require the widths of the confidence regions for the arm mean estimates to decrease at the same rate, which is not an assumption we can make in our setting (as the rates in our setting depend on properties of the estimator and its influence function).

# 5 Experiments

We studied the empirical sample complexities of the CS-LUCB and CS-SE algorithms for the task of selecting an identification formula for the causal DAG on the right, where the plate notation indicates that the path $X \leftarrow V_m \rightarrow Z_m \rightarrow Y$ is repeated $M$ times. Specifically, we considered the problem of selecting between the $2^M$ identification formulas obtained from the adjustment

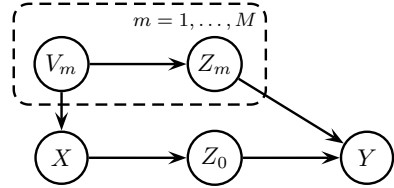

criterion using either $V_m$ or $Z_m$, for all $m \in [M]$, and the identification formula obtained from the frontdoor criterion using $Z_0$; i.e. between $2^M + 1$ different formulas. We compared our two algorithms against a simple uniform sampling baseline that samples every estimator the same number of times. The code implementing the experiments is available at `github.com/deepmind/abcei_mab`.

If we ignore cost, graphical criteria suggest that the adjustment formula with the lowest lower asymptotic variance is obtained by choosing $Z_m$ for all $m \in [M]$ (see Appendix A.2) . Therefore, we set the cost of including $Z_m$ or $V_m$ equal to 3 and 1, respectively ($X$ and $Y$ have no cost).

We assumed a binary exposure $X$ and considered causal contrast $\tau := \sum_x \lambda_x \mu_{\text{do}(x)}$ with $\lambda_0 = -1, \lambda_1 = 1$, corresponding to the average treatment effect. We used the AIPW estimator for the adjustment formulas and a Neyman orthogonal estimator derived from Fulcher et al. [6, Theorem 1] for the frontdoor formula (see Appendix A.3 for details); both have uncentered influence functions, thus the estimator from Eq. (2) was used.

## 5.1 Linear Model

In our first experiment, we considered observational data generated from the following linear model. For $m \in [M]$, $V_m \sim \mathcal{N}(0, I_2)$, $Z_m = A_m V_m + \epsilon_{z,m}$ for matrix $A_m \in \mathbb{R}^{3 \times 2}$ and $\epsilon_{z,m} \sim \mathcal{N}(0, .1 I_3)$, $X \sim \text{Bernoulli}\left(1 \Big/ \left(1 + e^{-\sum_{m=1}^M B_m^\top V_m}\right)\right)$ for vector $B_m \in \mathbb{R}^2$, and $Z_0$ is sampled from a categorical distribution conditioned on $X$. Specifically, for support points $\mathcal{S} := \{s_i \in \mathbb{R}^2 : i \in [10]\}$ and vectors $q_1$ and $q_0$ in the 9-simplex, we set $p(Z_0 = s_j | X = i) = q_i(j)$, where $q_i(j)$ is the $j$th element of $q_i$. Finally, $Y = \sum_{m=0}^M C_m^\top Z_m + \epsilon_y$ for vector $C_m \in \mathbb{R}^2$ and $\epsilon_y \sim \mathcal{N}(0, .1)$.

Specific observational distributions were obtained by sampling $U_m \sim \text{Uniform}[.1, .9]$, and then each element of $A_m$, $B_m$, and $C_m$, from $\mathcal{N}(0, U_m^2/4)$, $\mathcal{N}(0, U_m^2)$, and $\mathcal{N}(0, (2 - U_m)^2)$, respectively. The purpose of this sampling scheme was to produce distributions where the correlation between $V_m$ and $X$ and the correlation between $Z_m$ and $Y$ are not both large. This choice widens the gaps to the best estimator and results in a more interesting bandit problem. $B_0$ had elements independently sampled from $\mathcal{N}(0, 1/4)$.

To generate the distribution of $Z_0$, we sampled each support point of $\mathcal{S}$ from $\mathcal{N}(0, I_2)$, $G_1$ and $G_0$ from $\mathcal{N}(0, I_{10})$, and set $q_1 = \text{softmax}(.4G_1)$ and $q_0 = \text{softmax}(.4G_0)$. We then computed $\tau = \sum_{i=1}^{10}(q_1(i) - q_0(i))B_0^\top s_i$ and resampled $\mathcal{S}$, $q_0$, and $q_1$ until $\tau$ was larger than 1, which avoided the hardest instances and allowed us to run more simulations. As harder instances increase the sample complexity of the uniform algorithm the most, this resampling step does not inflate the advantages of CS-LUCB or CS-SE.

For the AIPW estimator, we used ridge regression and logistic regression to fit $\mu_x(\mathcal{Z}) := \mathbb{E}_p[Y | X = x, \mathcal{Z}]$ and $e_x(\mathcal{Z}) := p(X | \mathcal{Z})$ respectively. We used known confidence sequences for ridge regression (see Abbasi-Yadkori et al. [1, Theorem 2] or Lemma 7 in Appendix D), and confidence intervals for logistic regression were approximated by a standard central limit theorem (CLT) confidence interval, $\left[\hat{\sigma}^2 + z_{\alpha_n/2}\frac{std(\hat{\sigma}^2)}{\sqrt{n}}, \hat{\sigma}^2 + z_{1-\alpha_n/2}\frac{std(\hat{\sigma}^2)}{\sqrt{n}}\right]$, with $\delta = 0.1$ and $\alpha_n = 6\delta/\pi n^2$, following the construction described in Section 3.5.

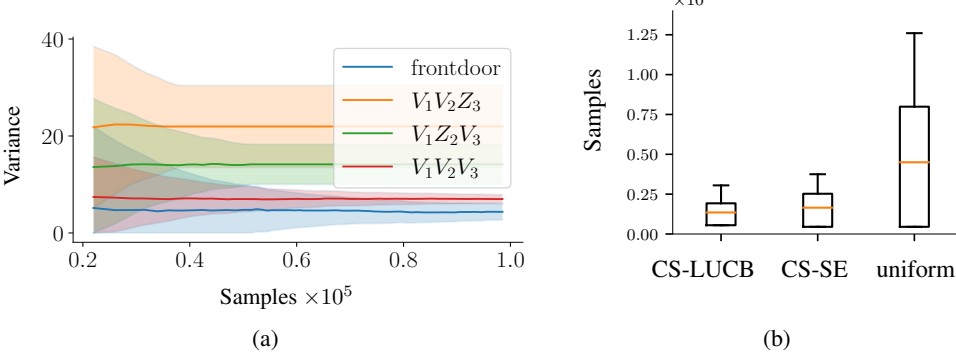

(a)                                                                (b)

Figure 1: (a): Example of confidence sequences corresponding to the four best estimators for CS-SE, up to the $8.1 * 10^5$ samples required to find the optimal estimator. (b): Box plot comparing sample complexities of CS-LUCB, CS-SE, and uniform sampling.

For the frontdoor estimator, we used ridge regression for $\mu_x(Z_0)$ and the MLE (i.e. empirical counts since the random variables are categorical) for $p(X = x)$ and $e_x(Z_0)$.

**Results.** For $M = 3$ (which produces 8 adjustment formulas and a frontdoor formula), we generated 10 observational distributions and, for each distribution, ran the algorithms 5 times from scratch on independent data. A typical run of CS-SE is shown in Fig. 1(a): confidence sequences of the four best formulas are plotted, and one can identify visually when sub-optimal estimators are removed from $S$. The algorithm terminates as soon as the two lowest confidence sequences no longer intersect. In Fig. 1(b), we provide a box plot for the distribution of sample complexities for the three algorithms. The orange line, box, and whiskers indicates the median, the IQR (the 25th percentile to the 75th percentile), and the outlier-filtered range (points with values $1.5 \times$IQR greater than the 75th percentile are removed), respectively. We see that some instances can be difficult for all algorithms (if, for example, many of the gaps are small), but CS-LUCB and CS-SE display substantial reduction in sample complexity w.r.t. the uniform sampling algorithm, on average needing 34% and 51% the number of samples to terminate.

## 5.2 Nonlinear Model

A key result from Theorems 1 and 2 is the second-order dependence on the rate of $\eta$ estimation, which allows the use of high-dimensional, nonparametric estimators for $\eta$. To showcase this, we considered a more complex nonlinear generation process. Given nonlinear functions $f_m, g_m, h_m$ for $m \in [M]$ and $g_0$, we generated observations as in Section 5.1, except with $X \sim$ Bernoulli $\left(1 \Big/ \left(1 + e^{-\sum_{m=1}^{M} h_m(V_m)}\right)\right)$, $Z_m = f_m(V_m) + \epsilon_{z,m}$, and $Y = \sum_{m=0}^{M} g_m(Z_m) + \epsilon_y$. We sampled the nonlinear functions to be element-wise from a Gaussian process prior with the RBF kernel on $10^d$ equidistant points in $(-2, 2)^d$, where $d$ is the dimension of the domain. We fit all the nonlinear functions with gradient-boosted regression trees with the same kernel. The confidence sequences for the nuisance functions were approximated using the same CLT-based construction used for logistic regression.

**Results.** Similar to the linear case, we generated 10 random observational distributions with 9 formulas and, for each distribution, ran each algorithm 5 times from scratch on independent data. We found a similar reduction in sample complexity as in the linear case (see Fig. 2(a)). We also explored how the sample complexity changes with the number of formulas. We sampled an observational distribution for all $M \in \{3, 4, 5, 6\}$ (corresponding to 9, 17, 33, and 65 formulas); for each distribution, we ran the algorithms from scratch 4 times on independent data and plotted the average sample complexities as a function of the number of formulas in Fig. 2(b). Our theory from Section 4 predicts that the sample complexity is determined not by the number of estimators but by the reciprocals of the squared gaps, which increases sub-linearly with the number of estimators as

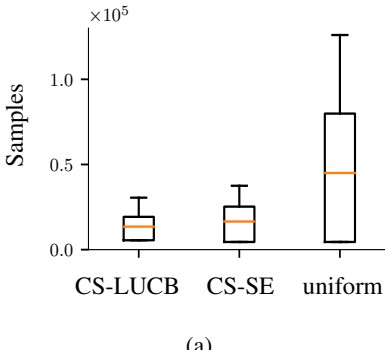 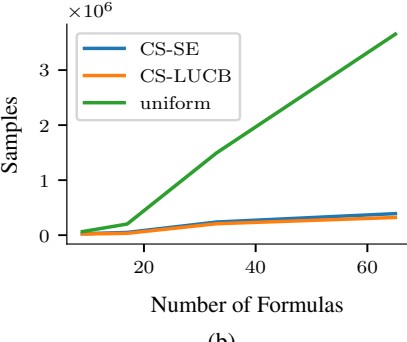

|(a)|(b)|

Figure 2: (a) Box plot of sample complexities with 9 formulas. (b) Sample complexities as a function of the number of formulas.

many of them have large gaps. In contrast, the sample complexity of the uniform algorithm should increase linearly. This prediction is confirmed by our experiments.

## 6 Discussion

Much of the literature on causal inference from observational data has focused either on *identifying* causal effects using structural knowledge of the causal graph underlying the data generation mechanism or on the design or selection of sample efficient *estimators*. The problem of selecting an identification formula using the efficiency of a corresponding estimator is only starting to receive attention.

Despite the recent progress of graphical criteria, comparing arbitrary formulas with statistical and practical considerations, such as cost or inability to observe certain covariates, remains an open problem. This work attempts to provide a functional answer to this problem: instead of trying to derive a solution from graph properties, we have provided a practical method that uses observational data to select a formula. When the graphical criteria do not apply, such as when the graph is partially unknown, latent variables exist, or when costs need to be considered, our methods can help the investigator reach a conclusion in a sample efficient way.

Our methods have the limitations of only guaranteeing an asymptotically optimal formula and relying on the availability of influence functions to quantify the asymptotic variance. To the best of our knowledge, more data-driven approaches to estimating the variance of estimators, such as resampling methods, do not have any finite-sample guarantees and are not appropriate for a bandit algorithm. Still, selecting between more general classes of estimators is an interesting direction.

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
