# A  Background on Causal Bayesian Networks

## A.1  Definitions

**Graph.** A *graph* $\mathcal{G}$ is a collection of nodes and links connecting pairs of nodes. Directed and undirected links give rise to *directed* and *undirected graphs*, respectively. A *path* from node $X_i$ to node $X_j$ is a sequence of linked nodes starting at $X_i$ and ending at $X_j$. A *directed path* is a path whose links are directed and pointing from preceding towards following nodes in the sequence.

**Directed Acyclic Graph (DAG).** A *DAG* is a directed graph with no directed paths starting and ending at the same node. A node $X_i$ with a direct link to $X_j$, $X_i \rightarrow X_j$, is called *parent* of $X_j$. In this case, $X_j$ is called *child* of $X_i$. A node is a *collider* on a path if it has two parents on that path. A node $X_i$ is an *ancestor* of a node $X_j$ if there exists a directed path from $X_i$ to $X_j$. In this case, $X_j$ is a *descendant* of $X_i$. We denote with $\text{de}(X_i, \mathcal{G})$ the set of *descendants*[2] of $X_i$ in $\mathcal{G}$.

**Bayesian Network.** A *Bayesian network* is a DAG in which nodes represent random variables and links express statistical relationships between the variables. Each node $X_i$ in the graph is associated with the conditional distribution $p(X_i|\text{pa}(X_i))$, where $\text{pa}(X_i)$ is the set of parents of $X_i$. The joint distribution of all nodes, $p(X_1, \ldots, X_I)$, is given by the product of all conditional distributions, i.e. $p(X_1, \ldots, X_I) = \prod_{i=1}^{I} p(X_i|\text{pa}(X_i))$.

A causal Bayesian network is a Bayesian network in which links represent causal influence rather than statistical dependence. $X$ is a *potential cause* of $Y$ if there exists a directed path, also called *causal path*, from $X$ to $Y$.

The sets of variables $\mathcal{X}$ and $\mathcal{Y}$ are *d-separated* by the set of variables $\mathcal{Z}$ in the DAG $\mathcal{G}$ (denoted as $\mathcal{X} \perp\!\!\!\perp_{\mathcal{G}} \mathcal{Y} \,|\, \mathcal{Z}$) if all paths from any element of $\mathcal{X}$ to any element of $\mathcal{Y}$ are *closed* (or *blocked*) by $\mathcal{Z}$. A path is *blocked* by $\mathcal{Z}$ if at least one of the following conditions is satisfied:

  (a)  There is a non-collider on the path which belongs to the conditioning set $\mathcal{Z}$.

  (b)  There is a collider on the path such that neither the collider nor any of its descendants belong to the conditioning set $\mathcal{Z}$.

If $\mathcal{X}$ and $\mathcal{Y}$ are d-separated by $\mathcal{Z}$, then $\mathcal{X}$ and $\mathcal{Y}$ are statistically independent given $\mathcal{Z}$ (denoted as $\mathcal{X} \perp\!\!\!\perp \mathcal{Y} \,|\, \mathcal{Z}$).

## A.2  Graphical Criteria for Selecting Asymptotically Optimal Adjustment Sets

In this section, we provide an overview of some of the recent results on adjustment set comparison and selection w.r.t. asymptotic variance using graphical criteria introduced in Henckel et al. [7], Rotnitzky and Smucler [27] and Smucler et al. [29].

**Forbidden Set.** The *forbidden set* relative to $(X, Y)$ in $\mathcal{G}$ is defined as $\text{forb}(X, Y, \mathcal{G}) = \text{de}(\text{cn}(X, Y, \mathcal{G}), \mathcal{G}) \cup X$, where $\text{cn}(X, Y, \mathcal{G})$ denotes the set of *causal nodes*, namely the set of nodes on causal paths from $X$ to $Y$, excluding $X$.

**Adjustment Set.** A (possibly empty) set $Z$ is an *adjustment set* relative to $(X, Y)$ if $p(y|\text{do}(x)) = p(y|x)$ if $Z = \emptyset$, and $p(y|\text{do}(x)) = \int_z p(y|x, z)p(z)dz$ otherwise. An adjustment set $Z$ is *minimal* if no subset of $Z$ is an adjustment set.

Adjustment sets can be read off from a given causal DAG using the *adjustment criterion* which generalizes the backdoor criterion [25, 26, 28].

**Adjustment Criterion.** The adjustment criterion requires that the following conditions hold: (a) $\text{forb}(X, Y, \mathcal{G}) \cap Z = \emptyset$ and (b) all non-causal paths from $X$ to $Y$ are blocked by $Z$.

The backdoor criterion and the adjustment criterion differ in the first condition (a). The backdoor criterion excludes all descendants of $X$, whilst the adjustment criterion excludes only the descendants of the causal nodes relative to $(X, Y)$.

Below we give two lemmas for a DAG $\mathcal{G}$ with node set $V$, and $X \subset V$, $Y \in V \setminus X$ with $X$ a random vector taking values on a finite set. We use, e.g., $\sigma_B^2$ to indicate the asymptotic variance

---

[2]Following the convention that a node is an ancestor and descendant of itself.

of an asymptotically linear estimator $\hat{\mu}_{\text{do}(x)}$ of $\mu_{\text{do}(x)} := \mathbb{E}[Y|\text{do}(X = x)]$ based on the adjustment criterion using that uses adjustment set $B$.

**Lemma 2.** *(Addition of precision set G) Suppose that $B \subset V \setminus \{X, Y\}$ is an adjustment set relative to $(X, Y)$ in $\mathcal{G}$, and that $G$ is a disjoint set with $B$ such that $X \perp\!\!\!\perp_{\mathcal{G}} G \,|\, B$. Then $(G, B)$ is also an adjustment set relative to $(X, Y)$ in $\mathcal{G}$ and $\sigma_B^2 - \sigma_{G \cup B}^2 \geq 0$.*

**Lemma 3.** *(Removal of overadjustement set B) Suppose that $(G \cup B) \subset V \setminus \{X, Y\}$ is an adjustment set relative to $(X, Y)$ in $\mathcal{G}$, with $G \cap B = \emptyset$, and that $Y \perp\!\!\!\perp_{\mathcal{G}} B \,|\, G, X$. Then $G$ is also an adjustment set relative to $(X, Y)$ in $\mathcal{G}$ and $\sigma_{G \cup B}^2 - \sigma_G^2 \geq 0$.*

Lemma 2 quantifies the reduction in variance associated with supplementing an adjustment set $B$ with a *precision set* $G$, i.e with variables that are independent of $X$ when conditioning on $B$. Lemma 3 quantifies the increase in variance incurred by keeping an *overadjustment set* $B$, i.e. variables that are associated with $X$ but do not help predicting $Y$ given the remaining adjusting variables $G$ and $X$. In general $B$ is more harmful the weaker the association between $Y$ and $G$ within levels of $X$.

**Theorem 4.** *Suppose that $G \subset V \setminus \{X, Y\}$ and $B \subset V \setminus \{X, Y\}$ are adjustment set relative to $(X, Y)$ in $\mathcal{G}$ such that $X \perp\!\!\!\perp_{\mathcal{G}} G \setminus B \,|\, B$ and $Y \perp\!\!\!\perp_{\mathcal{G}} B \setminus G \,|\, G, X$. Then $\sigma_B^2 - \sigma_G^2 \geq 0$.*

The proof follows from the equivalence $\sigma_B^2 - \sigma_G^2 = \sigma_B^2 - \sigma_{B \cup (G \setminus B)}^2 + \sigma_{G \cup (B \setminus G)}^2 - \sigma_G^2$, and from applying Lemma 2 to $\sigma_B^2 - \sigma_{B \cup (G \setminus B)}^2$ (supplementation with precision set $G \setminus B$) and Lemma 3 to $\sigma_{G \cup (B \setminus G)}^2 - \sigma_G^2$ (deletion of overadjustment set $B \setminus G$).

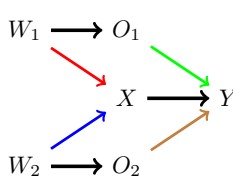

Not all pairs of valid adjustment sets can be ordered using Theorem 4. In the DAG on the left, $Z = \{O_1, W_2\}$ and $\tilde{Z} = \{O_2, W_1\}$ are adjustment sets relative to $(X, Y)$ which cannot be compared as $X \not\perp\!\!\!\perp_{\mathcal{G}} W_1 \,|\, Z$ nor $X \not\perp\!\!\!\perp_{\mathcal{G}} W_2 \,|\, \tilde{Z}$. The set $Z$ yields a smaller asymptotic variance than $\tilde{Z}$ if the association encoded in the green link is stronger than that in the brown link and the one encoded in the blue link is weaker than the one in the red link.

If an adjustment set relative to $(X, Y)$ in $\mathcal{G}$ exists $G = O(X, Y, \mathcal{G}) = \text{pa}(\text{cn}(X, Y, \mathcal{G}), \mathcal{G}) \setminus \text{forb}(X, Y, \mathcal{G})$ is an adjustment set that satisfies the independence conditions of Theorem 4 with respect to any adjustment set $B$. Therefore, we have the following important corollary to Theorem 4.

**Theorem 5.** *If an adjustment set $Z$ relative to $(X, Y)$ in $\mathcal{G}$ exists then $\sigma_Z^2 - \sigma_O^2 \geq 0$ ($O := O(X, Y, \mathcal{G})$). $O$ is called asymptotically optimal adjustment set.*

### A.3 A Neyman Orthogonal Frontdoor Criterion Estimator

We can obtain an estimator $\hat{\mu}_{\text{do}(x)}$ of $\mu_{\text{do}(x)} := \mathbb{E}[Y|\text{do}(X = x)]$ based on the frontdoor criterion

$$\mu_{\text{do}(x)} = \sum_z p(\mathcal{Z} = z|X = x) \sum_{x'} p(X = x') \mu_{x'}(\mathcal{Z} = z)$$

in a variety of ways. For example, if the distributions are all parametric with parameter $\theta$ (e.g. if they are tabular), then we may obtain an asymptotically linear estimate for $\hat{\mu}$ by estimating $\hat{\theta}$ and using the plug-in estimate

$$\hat{\mu}_{\text{do}(x)} = \sum_z p(\mathcal{Z} = z|X = x, \hat{\theta}) \sum_{x'} p(X = x'|\hat{\theta}) \mu_{x'}(\mathcal{Z} = z|\hat{\theta}).$$

However, if we wish to use non-parametric estimators for the nuisance function, we need to look for semi-parametric estimators with Neyman orthogonality [2]. The recent work of Jung et al. [15] provides algorithms for constructing such estimators and even guarantees that the influence functions are uncentered. For a generalized front-door criterion where $X$, $\mathcal{Z}$, and $Y$ are allowed to share a confounding parent $C$, Fulcher et al. [6, Theorem 1] derives the efficient influence function. The case

corresponding to the front-door criteria and binary $X$ is

$$
\psi(W, \eta) = (Y - \mathbb{E}_p[Y|\mathcal{Z}, X]) \left( \frac{p(\mathcal{Z}|X=1)}{p(\mathcal{Z}|X)} - \frac{p(\mathcal{Z}|X=0)}{p(\mathcal{Z}|X)} \right)
$$
$$
+ \sum_{z'} \mathbb{E}_p[Y|\mathcal{Z}=z', X] \left( p(\mathcal{Z}=z'|X=1) - p(z'|X=0) \right)
$$
$$
+ \left( \frac{X}{p(X=1)} - \frac{1-X}{p(X=0)} \right) \sum_{x'} \mathbb{E}_p[Y|\mathcal{Z}, X=x']p(X=x')
$$
$$
- \left( \frac{X}{p(X=1)} - \frac{1-X}{p(X=0)} \right) \sum_{x', z'} \mathbb{E}_p[Y|\mathcal{Z}=z', X=x']p(\mathcal{Z}=z'|X)p(X=x'),
$$

which is the formula implemented in the experiments. The nuisance function is $\eta = (p(X = x), p(X = x|Z = z), \mathbb{E}_p[Y|Z = z, X = x])$.

## B  Background on Bandit Algorithms

Multi-armed bandits are a popular and successful framework for modeling sequential decision making problems under uncertainty. Like most of the Online Learning literature, a bandit problem is phrased in the language of repeated games between the learner (or investigator) and the environment. Generally, the learner has $K$ actions to choose from. The game proceeds in rounds $t = 1, 2, \ldots$, and during each round, the learner chooses arm $k_t$, the environment chooses a loss function $\ell_t \in \{1, \ldots, K\} \to \mathbb{R}$, and the learner observes $\ell_t(k_t)$ but not $\ell_t$ at any other possible choice. It is this limitation on the feedback that makes the problem interesting.

By now, several great textbooks exist on multi-armed bandits [20], so we will only focus on explaining the background specific for best-arm identification. Multi-armed bandits usually refers to the regret-minimization formulation where the learner chooses actions and tries to incur cumulative loss almost as small as the cumulative loss of the best single fixed action (this quantity is known as the regret). In contrast, the best-arm-identification problem is to identify the arm with the smallest average loss without regard to any losses accumulated along the way. Intuitively, a best-arm-identification algorithm will explore more than a regret-minimization algorithm, since it is not deterred from playing costly arms that could still provide information.

There are two goals explored in the literature. In fixed-confidence, the learner wishes to find the best arm with a guaranteed minimum confidence in as few samples as possible; analyzing algorithms in this setting usually requires showing a proof of correctness (that the probability of correctness is sufficiently high) and a bound on the sample complexity (since the sample complexity is random, this is usually shown in expectation or with high probability). In the fixed-budget setting, the learner has a fixed number of samples that can be collected and wishes to maximize the probability of finding the correct arm. Our setting more naturally fits into the fixed-confidence setting.

The best-arm-identification protocol is the following.

---

**Experimental Protocol**

---

**Given:** Number of actions $K$, desired confidence $\delta \in (0, 1)$, error tolerance $\epsilon > 0$
**Fixed:** loss distributions $\nu_1, \ldots, \nu_K$ with means $\mu_1, \ldots, \mu_K$
**for** $t = 1, 2, \ldots, T$: **do**

> The learner chooses action $k_t \in [K]$
> The learner observes $Y_t \sim \nu_{k_t}$
> The learner decides whether to stop

**end**
The goal is to return an $(\epsilon, \delta)$-PAC index $\hat{k}$ with

$$
\mathbb{P}\left(\mu_{\hat{k}} \geq \min_k \mu_k + \epsilon\right) \leq \delta.
$$

---

In words, the learner may sample from one of $K$ distributions every round and wants to identify the distribution with the smallest mean. More specifically, the learner wants to guarantee that the returned index corresponds to a mean that is at most $\epsilon$ suboptimal with probability at least $1 - \delta$, i.e. one that is probably approximately correct (PAC).

Typically, an algorithm is analyzed by proving an upper bound on the sample complexity in terms of some parameters that describe the difficulty of the particular instance. These upper bounds often match, at least up to log factors, lower bounds on sample complexity that hold for any PAC algorithm. The instance-dependent parameters are typically the sub-optimality gap of the means, defined as $\Delta_k := \mu_k - \mu_{k^*}$, where $k^* = \arg\min_k \mu_k$.

The most common bandit algorithms in the literature fall into a few categories.

1. Action elimination algorithms (e.g. Successive Elimination [4]) keep a set of arms that could be optimal. They alternate between sampling all the arms in this set and using confidence intervals to prune the set.

2. Optimistic algorithms, like UCB [20] and LUCB [17], first construct the most optimistic problem instance every round that is consistent with the confidence intervals (e.g. by assuming the smallest mean values in the confidence sets), then act greedily as if this instance was true.

3. Track-and-Stop-style [19] algorithms, which require that the distributions live in a parametric family, compute an asymptotic lower bound on the sample complexity and try to keep the empirical sampling proportion close to the sampling proportion achieving the lower bound.

When $\epsilon = 0$, the sample complexity lower bound is

$$\Omega\left(\sum_{k \neq k^*} \Delta_k^{-2} \log(\log(\Delta_k^{-2})/\delta)\right).$$

There exist algorithms with matching upper bounds, such as lil'UCB and Exponential Gap Elimination [12], but these algorithms rely on the fact that the confidence interval widths all decrease at a deterministic, Hoeffding-like rate (i.e. of $\mathcal{O}(n^{-1/2}\log(1/\delta))$), and are not generally suitable for our setting. We choose to adapt versions of Successive Elimination and LUCB, which work without modification for general confidence interval widths, and leave extending these optimal algorithms as future work.

When $\epsilon > 0$, the most cited lower bound is that of Mannor and Tsitsiklis [21], which is

$$\Omega\left(\frac{K}{\epsilon^2} \log(1/\delta)\right).$$

To the best of our knowledge, a lower bound that matches the terms in our analysis, $\frac{\Delta_k}{2} \vee \epsilon$, does not exist. However, the lower bounds bound the literature are, in general, not useful for our setting; they tend to assume that the arm rewards have well controlled tail behavior (e.g. are sub-Gaussian or belong in some exponential family). In contrast, we do not wish to make such assumptions about the confidence interval widths. As such, any instance-dependent lower bounds would need to depend on the widths for all the estimators, significantly complicating the analysis.

## B.1 The general LUCB algorithm

The version of the LUCB algorithm given in the main text is tailored for best-arm identification. The original algorithm [17] is written for $m$-best-arm identification and can also be adapted for our estimator-selection setting; it is presented below.

**Algorithm 3** top-$m$ CS-LUCB

---

**Input** $B > 1, \tilde{\tau} > 0, m \geq 1$
$\{\psi_k, \hat{\eta}_k, u_k, c_k : k \in [K]\}$
**for** *k=1,..., K* **do**
    Obtain $B$ new samples $\mathcal{D}$
    Add half of $\mathcal{D}$ to $\mathcal{D}_k^\eta$ and half to $\mathcal{D}_k^\sigma$
    $\hat{\sigma}_k^2, \beta_k \leftarrow$ CSUpdate$(u_k, \hat{\eta}_k, \psi_k, L, \tilde{\tau}, \mathcal{D}_k^\eta, \mathcal{D}_k^\sigma)$
**end**
**for** $t = 1, 2, \ldots$ **do**
    $K_t \leftarrow \arg\min_{K \subseteq [K]:|K|=m} \sum_{k \in K} c_k \hat{\sigma}_k^2$
    $l_t \leftarrow \arg\max_{k \in K_t} c_k(\hat{\sigma}_k^2 + \beta_k)$
    $u_t \leftarrow \arg\min_{k \notin K_t} c_k(\hat{\sigma}_k^2 - \beta_k)$
    **if** $c_{l_t}\left(\hat{\sigma}_{l_t}^2 + \beta_{l_t}\right) \leq c_{u_t}\left(\hat{\sigma}_{u_t}^2 - \beta_{u_t}\right) - \epsilon$ **then**
        Return $K_t$
    **end**
    **for** $k \in u_t, l_t$ **do**
        Obtain $B$ new samples $\mathcal{D}$
        Add half of $\mathcal{D}$ to $\mathcal{D}_k^\eta$ and half to $\mathcal{D}_k^\sigma$
        $\hat{\sigma}_k^2, \beta_k \leftarrow$ CSUpdate$(u_k, \hat{\eta}_k, \psi_k, L, \tilde{\tau}, \mathcal{D}_k^\eta, \mathcal{D}_k^\sigma)$
    **end**
**end**

---

## C   Confidence Sequences

We begin with a "canonical assumption" used throughout the literature (see e.g. Howard et al. [9]). The use of $\psi$ is this section is completely separate from the use of $\psi$ for an uncentered influence function in the main paper; the meaning of $\psi$ is changed to aid the reader in appealing to the confidence sequence literature. For a random processes $(S_t)_{t>0} \in \mathbb{R}$ and $(V_t)_{t>0} \in \mathbb{R}$ and function $\psi : \mathbb{R} \to \mathbb{R}$, we say that $(S_t)_{t>0}$ is sub-$\psi$ with variance process $V_t$ if, for all $t > 0$,

$$\mathbb{E}_{t-1}[\exp\{\lambda S_t - \psi(\lambda, V_t)\}] \leq \exp\{\lambda S_{t-1} - \psi(\lambda, V_{t-1})\} \tag{3}$$

for all $\lambda$ in some subset of $\mathbb{R}$ and where $\mathbb{E}_t$ is the conditional expectation w.r.t. $\mathcal{F}_t :=$ $\sigma(S_1, \ldots, S_t, V_1, \ldots, V_t)$. Equivalently, we require that $\exp\{\lambda^\top S_t - \psi(\lambda, V_t)\}$ is a nonnegative supermartingale adapted to $\mathcal{F}_t$. Unless otherwise specified, we assume that Eq. (3) holds for all $\lambda \in \mathbb{R}^d$. Intuitively, the function $\psi$ controls the tail behavior. We extend this definition to vector-valued random processes $(S_t)_{t>0} \in \mathbb{R}^d$ with variances process $(V_t)_{t>0} \in \mathbb{R}^{d \times d}$ by requiring Eq. (3) holds for all $\lambda \in \mathbb{R}^d$ and replacing $\lambda S_t$ with $\lambda^\top S_t$ ($\psi$ is still real-valued).

The definitions for a sub-exponential random variable and sub-$\psi$ process with sub-exponential boundary are very related and summarized below.

**Definition 4.** *A random variable $X \in \mathbb{R}$ with mean $\mu$ is $\nu$ sub-exponential if there exist constants $\nu$ and $c$ such that*

$$\mathbb{E}\left[e^{\lambda(X-\mu)}\right] \leq e^{\frac{\nu\lambda^2}{2}} \ \forall \ \lambda \in [0, 1/c).$$

*A random process $S_t$ is $\nu$ sub-exponential with scale $c > 0$ and variance process $V_t$ if the "canonical assumption" is satisfied for*

$$\psi_{E,c}(\lambda) = \nu \frac{-\log(1 - c\lambda) - c\lambda}{c^2} \ \forall \ \lambda \in [0, 1/c). \tag{4}$$

By Howard et al. [9, Proposition 5], these two conditions are equivalent.

The sub-exponential property is useful for bounding the variance; roughly, the square of a sub-Gaussian random variable is sub-exponential. Specific parameter values may be obtained by comparing moments. For example, [8, Appendix B] uses this approach to show that, if $X$ is $\sigma^2$ sub-Gaussian, then

$$\mathbb{E}\left[e^{\lambda(X^2 - \mathbb{E}[X^2])}\right] \leq e^{16\lambda^2 \sigma^4} \ \ \forall \lambda \in [0, (4\sigma^2)^{-1}),$$

which satisfies the definition of sub-Exponential for the values in Lemma 1, repeated below.

**Lemma 1.** *If $X$ is $\sigma^2$ sub-Gaussian, then $X^2$ is a sub-exponential random variable with $\nu = 4\sqrt{2}\sigma^4$ and $c = 4\sigma^2$.*

The purpose of stating the tail control in terms of sub-$\psi$ assumptions is to enable the reader to quickly adapt confidence sequence results from the literature. For example, we can obtain a confidence sequence on a bounded random variable (which is useful for obtaining a confidence sequence for tabular conditional probability distributions) using Howard et al. [10, Proposition 11], which implies that confidence sequences for bounded random variables are sub-Gaussian.

Next, we present a general purpose confidence sequence for averages of sub-Gaussian and sub-exponential random variables that is used in the construction of Corollary 1. This confidence sequence is a special case of [10, Equation 8], which derives a boundary for what the authors call "sub-Gamma" random variables. Their Proposition 11 implies that this boundary holds for sub-Gaussian and sub-exponential random variables as well.

**Lemma 4.** *Let $\gamma > 0$ and $m > 0$ be scalar parameters, and let $h : \mathbb{R}_{\geq 0} \to \mathbb{R}_{\geq 0}$ be an increasing function with summable reciprocals. With $\ell(v) := \log(h(\log_\gamma(v/m)) + \log(2/\alpha)$, define*

$$u_n(a, b) = \frac{\gamma^{1/4} + \gamma^{-1/4}}{\sqrt{2}n}\sqrt{(an \vee m)\ell(an \vee m)} + b\frac{\sqrt{\gamma} + 1}{n}\ell(an \vee m).$$

*If $W$ is a $\lambda$ sub-exponential random variable with scale $c$, then $\{u_n(\lambda, c)\}$ is a boundary sequence at level $\alpha$ for $|\mathbb{E}_n[W] - \mathbb{E}[W]|$. If $W$ is a $\lambda$ sub-Gaussian random variable, then $u_n(\lambda, 0)$ is a boundary sequence at level $\alpha$ for $|\mathbb{E}_n[W] - \mathbb{E}[W]|$.*

We can now show a result for the case when $\psi(W, \eta)$, at the true $\eta$, is $\lambda$ sub-Gaussian by combining Theorem 1 with Lemma 4 and Lemma 1. This Lemma is a more general version of Corollary 1 in the main text.

**Corollary 2.** *Let $\alpha \in (0, 1)$ and assume the same setting as Theorem 1, and additionally that $\psi(W, \eta)$ is $\lambda$ sub-Gaussian. Then, for $\{u_n(a, b)\}$ and $\{u_n(a)\}$ as defined in Lemma 4 and $n' = |\mathcal{D}_n^\sigma|$,*

$$\mathbb{P}\left(\exists n \geq 1 : \left|\hat{\sigma}^2(\mathcal{D}_n) - \sigma^2\right| \geq 2L^2(u_n^\eta)^2 + u_{n'}(8\lambda^2, 2\lambda) + u_{n'}^2(\lambda) + 2\tilde{\tau}u_{n'}(\lambda)\right) \leq \alpha.$$

*In particular, let $m > 0$ and define $\lambda' = \lambda \vee 8\lambda^2$ and $n' := (18.6\lambda \log(\lambda n/m) + \log(2/\alpha)) \vee m/\lambda'$. Then, the confidence sequence with parameters $\gamma = 2$ and $h(k) = 2^{2k+1}$ (as suggested by [10]) evaluates to*

$$\mathbb{P}\left(\exists n \geq n' : \left|\hat{\sigma}^2(\mathcal{D}_n) - \sigma^2\right| \geq 2L^2(u_n^\eta)^2 + \frac{5\left(\sqrt{2\lambda} + \tilde{\tau}\right)}{8}\sqrt{\frac{1}{n}\left(2\lambda \log(\lambda' n/m) + \log\frac{2}{\alpha}\right)}\right) \leq \alpha.$$

# D   Confidence Sequences for Parameters

As an illustration that confidence sequences for fit parameters are possible, we will reproduce the confidence sequence for the ordinary least squares estimator from Abbasi-Yadkori et al. [1, Theorem 2]. Assume the linear model

$$Y_t = z_t^\top \beta + \epsilon_t, \quad t = 1, 2, \dots, \tag{5}$$

where $z_t$ are arbitrary vectors and $\epsilon_t$ are i.i.d., zero mean random variables. The errors are known to be a sub-Gaussian stochastic process, meaning that we can derive a sub-$\psi$ condition for them. The claim, shown in e.g. [3], is repeated below in our notation.

**Lemma 5.** *Let $S_t = \sum_{s=1}^t z_s \epsilon_s$ and assume the linear model Eq. (5). If $\epsilon_t$ is $\sigma^2$ sub-Gaussian, then $(S_t)_{t>0}$ is sub-$\psi$ with $\psi(\lambda, V) = \frac{\sigma^2}{2}\lambda^2 V$ and variance process $V_t = \sum_{s=1}^t z_s z_s^\top$.*

An interesting feature of this confidence sequence is that it is self-normalized; the bound on $S_t$ is a function of the data as it depends on $V_t$. Using the boundary sequence derived in [1], we can show the following.

**Lemma 6.** *Let $S_t = \sum_{s=1}^t z_s \epsilon_s$ and let $D_t$ be the data matrix (i.e. the matrix with rows $z_1, \dots, z_t$. Assume the linear model Eq. (5), and let and $\rho > 0$. If $\epsilon_t$ is $\lambda$ sub-Gaussian, then $(S_t)_{t>0}$ has*

$$P\left(\forall t > 0 : S_t^\top (D_t^\top D_t + \sigma^{-2}\rho I)^{-1} S_t \leq \lambda \log\left(\frac{1}{\alpha^2}\frac{\det(\lambda D_t^\top D_t + \rho I)}{\rho^d}\right)\right) \geq 1 - \alpha.$$

*In the context of estimating a nuisance function $\eta$, we have $\hat{\beta}_t - \beta = (D_t^\top D_t)^{-1} S_t$, so*

$$u_n = \sqrt{\lambda \log\left(\frac{1}{\alpha^2} \frac{\det(\lambda D_t^\top D_t + \rho I)}{\rho^d}\right)}$$

*is a boundary sequence of level $\alpha$ on the random variable*

$$(\hat{\beta}_t - \beta)^\top (D_t^\top D_t)(D_t^\top D_t + \sigma^{-2}\rho I)^{-1}(D_t^\top D_t)(\hat{\beta}_t - \beta).$$

# E   Delayed Proofs

*Proof of Theorem 1.* We use the shorthand $\hat{\eta}_n := \hat{\eta}(\mathcal{D}_n^\eta)$. Also, let $\mathbb{E}_n$ denote the empirical expectation w.r.t. $\mathcal{D}_n^\sigma$. The sample splitting estimator defined in Eq. (2) splits the data into two folds, uses the first fold to evaluate $\hat{\eta}_n$, and the second fold to calculate

$$\mathrm{var}_n[\psi(W, \hat{\eta}_n)] = \mathbb{E}_n[(\psi(W, \hat{\eta}_n) - \mathbb{E}_n[\psi(W, \hat{\eta}_n)])^2].$$

Using the identity $\mathrm{var}_n[\psi(W, \hat{\eta}_n)] = \mathbb{E}_n[\psi(W, \hat{\eta}_n)^2] - \mathbb{E}_n[\psi(W, \hat{\eta}_n)]^2$, we can expand $\hat{\sigma}^2(\mathcal{D}_n) - \sigma^2$ as

$$
\begin{aligned}
\hat{\sigma}^2(\mathcal{D}_n) - \sigma^2 &= \mathrm{var}_n[\psi(W, \hat{\eta}_n)] - \mathrm{var}[\psi(W, \eta)] \\
&= \mathbb{E}_n[\psi^2(W, \hat{\eta}_n)] - \mathbb{E}[\psi^2(W, \eta)] - \mathbb{E}_n[\psi(W, \hat{\eta}_n)]^2 + \mathbb{E}[\psi(W, \eta)]^2 \\
&= \mathbb{E}_n[\psi^2(W, \hat{\eta}_n)] - \mathbb{E}_n[\psi^2(W, \eta)] - \mathbb{E}_n[\psi(W, \hat{\eta}_n)]^2 + \mathbb{E}_n[\psi(W, \eta)]^2 \\
&\quad + \left(\mathbb{E}_n[\psi^2(W, \eta)] - \mathbb{E}[\psi^2(W, \eta)]\right) + \left(\mathbb{E}_n[\psi(W, \eta)]^2 - \mathbb{E}[\psi(W, \eta)]^2\right) \\
&= \mathbb{E}_n\left[(\psi(W, \hat{\eta}_n) - \psi(W, \eta))^2\right] + 2\mathbb{E}_n\left[\psi(W, \eta)(\psi(W, \hat{\eta}_n) - \psi(W, \eta))\right] \quad (6) \\
&\quad - \mathbb{E}_n[\psi(W, \hat{\eta}_n) - \psi(W, \eta)]\mathbb{E}_n[\psi(W, \hat{\eta}_n) + \psi(W, \eta)] \\
&\quad + \left(\mathbb{E}_n[\psi^2(W, \eta)] - \mathbb{E}[\psi^2(W, \eta)]\right) + \left(\mathbb{E}_n[\psi(W, \eta)]^2 - \mathbb{E}[\psi(W, \eta)]^2\right).
\end{aligned}
$$

Using the L-Lipschitz property of $\psi$, we can bound the first term by

$$\left|\mathbb{E}_n\left[(\psi(W, \hat{\eta}_n) - \psi(W, \eta))^2\right]\right| \leq L^2 \|\hat{\eta}_n - \eta\|^2. \tag{7}$$

Rearranging the second and third terms and using Cauchy-Schwarz, we have

$$
\begin{aligned}
&|2\mathbb{E}_n\left[\psi(W, \eta)(\psi(W, \hat{\eta}_n) - \psi(W, \eta))\right] - \mathbb{E}_n[\psi(W, \hat{\eta}_n) - \psi(W, \eta)]\mathbb{E}_n[\psi(W, \hat{\eta}_n) + \psi(W, \eta)]| \\
&= |\mathbb{E}_n\left[(2\psi(W, \eta) - \mathbb{E}_n[\psi(W, \hat{\eta}_n) + \psi(W, \eta)])(\psi(W, \hat{\eta}_n) - \psi(W, \eta))\right]| \\
&\leq \sqrt{\mathbb{E}_n\left[(2\psi(W, \eta) - \mathbb{E}_n[\psi(W, \hat{\eta}_n) + \psi(W, \eta)])^2\right]} \sqrt{\mathbb{E}_n\left[(\psi(W, \hat{\eta}_n) - \psi(W, \eta))^2\right]} \\
&= \sqrt{\mathbb{E}_n\left[(\psi(W, \eta) - \psi(W, \hat{\eta}_n))^2\right]} \sqrt{\mathbb{E}_n\left[(\psi(W, \hat{\eta}_n) - \psi(W, \eta))^2\right]} \\
&= \mathbb{E}_n\left[(\psi(W, \hat{\eta}_n) - \psi(W, \eta))^2\right] \\
&\leq L^2 \|\hat{\eta}_n - \eta\|^2.
\end{aligned}
$$

Using the boundary sequence at $n$, $u_n^{(\psi,2)}$, to control the forth term is immediate. We can tackle the fifth and final term by writing

$$
\begin{aligned}
\left|\mathbb{E}_n[\psi(W, \eta)]^2 - \mathbb{E}[\psi(W, \eta)]^2\right| &\leq \left|\left(\mathbb{E}[\psi(W, \eta)] - u_n^{(\psi,1)}\right)^2 - \mathbb{E}[\psi(W, \eta)]^2\right| \\
&\leq \left(u_n^{(\psi,1)}\right)^2 + 2\left|u_n^{(\psi,1)}\mathbb{E}[\psi(W, \eta)]\right| \\
&\leq \left(u_n^{(\psi,1)}\right)^2 + 2|\tau|u_n^{(\psi,1)} \leq \left(u_n^{(\psi,1)}\right)^2 + 2\tilde{\tau}u_n^{(\psi,1)}.
\end{aligned}
$$

Collecting the terms above will yield the theorem statement. $\qquad\square$

*Proof of Corollary 1.* Since $\psi(W, \eta)$ is assumed to be $\lambda$ sub-Gaussian, Lemma 1 implies that $\psi^2(W, \eta)$ is $4\sqrt{2}\lambda^2$ sub-exponential with scale $c = 2\lambda$. Hence, Lemma 4 guarantees that

$$u_n^{(\psi,1)} = u_n(\lambda, 0) \text{ and } u_n^{(\psi,2)} = u_n(4\sqrt{2}\lambda^2, 4\lambda),$$

with the $u_n(\lambda, c)$ defined in Lemma 4, are boundary sequences at level $\alpha$ for $|\mathbb{E}_{\mathcal{D}_n}[\psi(W, \eta)] - \mathbb{E}[\psi(W, \eta)]|$ and $|\mathbb{E}_{\mathcal{D}_n}[\psi(W, \eta)]^2 - \mathbb{E}[\psi(W, \eta)]^2|$, respectively. Substituting these expressions into Theorem 1 yields

$$\mathbb{P}\left(\forall n \geq 1 \left|\hat{\sigma}^2(\mathcal{D}_n) - \sigma^2\right| \leq 2L^2 \left\|\hat{\eta}(\mathcal{D}_n^\eta) - \eta\right\|^2 + u_n(4\sqrt{2}\lambda^2, 4\lambda) + (u_n(\lambda, 0))^2 + 2\tilde{\tau}u_n(\lambda, 0)\right) \geq 1 - 3\alpha.$$

We can glean some intuition by expanding the boundary functions as

$$u_n(4\sqrt{2}\lambda^2, 4\lambda) + u_n(\lambda, 0)^2 + 2\tilde{\tau}u_n(\lambda, 0) = \frac{\gamma^{1/4} + \gamma^{-1/4}}{\sqrt{2n}}\sqrt{(4\sqrt{2}\lambda^2 n \vee m)\ell(4\sqrt{2}\lambda^2 n \vee m)}$$

$$+ 4\lambda\frac{\sqrt{\gamma} + 1}{n}\ell(4\sqrt{2}\lambda^2 n \vee m)$$

$$+ \frac{(\gamma^{1/4} + \gamma^{-1/4})^2}{2n^2}(\lambda n \vee m)\ell(\lambda n \vee m)$$

$$+ 2\tilde{\tau}\frac{\gamma^{1/4} + \gamma^{-1/4}}{\sqrt{2n}}\sqrt{(\lambda n \vee m)\ell(\lambda n \vee m)}.$$

Letting $\lambda' = \lambda \vee 4\sqrt{2}\lambda^2$ and assuming that $n \geq m/\lambda'$, we can simplify to

$$u_n(4\sqrt{2}\lambda^2, 4\lambda) + u_n(\lambda, 0)^2 + 2\tilde{\tau}u_n(\lambda, 0) \leq \frac{\lambda\ell(\lambda'n)}{n}\left(4(\sqrt{\gamma} + 1) + \frac{(\gamma^{1/4} + \gamma^{-1/4})^2}{2}\right)$$

$$+ (1 + 2\tilde{\tau})\frac{\gamma^{1/4} + \gamma^{-1/4}}{\sqrt{2n}}\sqrt{\lambda'\ell(\lambda'n)}.$$

The second term will dominate quickly because it scales with $1/\sqrt{n}$. Hence, for

$$\sqrt{n} \geq \frac{\sqrt{\lambda'\ell(\lambda'n)}}{\sqrt{2}(1 + 2\tilde{\tau})}\left(\frac{8(\sqrt{\gamma} + 1)}{\gamma^{1/4} + \gamma^{-1/4}} + \gamma^{1/4} + \gamma^{-1/4}\right) \vee \sqrt{\frac{m}{\lambda'}},$$

we have

$$u_n(4\sqrt{2}\lambda^2, 4\lambda) + u_n(\lambda, 0)^2 + 2\tilde{\tau}u_n(\lambda, 0) \leq 2(1 + 2\tilde{\tau})\frac{\gamma^{1/4} + \gamma^{-1/4}}{\sqrt{2n}}\sqrt{\lambda'\ell(\lambda'n)} = \frac{3 + 6\tilde{\tau}}{\sqrt{n}}\sqrt{\lambda'\ell(\lambda'n)},$$

where the last equality was from the simple choice $\gamma = 4$. With this value of $\gamma$ and noting that $(1 + 2\tilde{\tau})^{-1} \leq 1$, a direct calculation shows that it suffices to take the lower bound on $n$ to be

$$n \geq 91\lambda'\ell(\lambda'n) \vee \frac{m}{\lambda'}.$$

Coming even closer to a concrete bound, we evaluate the bound for $h(k) = \frac{\eta^{sk}}{1 - \eta^{-s}}$, as suggested by Howard et al. [10]. Then, choosing $s = 2$, $\gamma = 4$ ensuring that $h(k) = 2^{k+1}$, we have

$$\ell(v) = \log\left(h(\log_4(v/m))\right) + \log(1/\alpha) = \left(\log_4\left(\frac{v}{m}\right) + 1\right)\log(2) + \log(1/\alpha)$$

$$\leq \frac{1}{2}\log(v/m) + \log(2) + \log(1/\alpha).$$

The total bound is then

$$u_n(4\sqrt{2}\lambda^2, 4\lambda) + u_n(\lambda, 0)^2 + 2\tau u_n(\lambda, 0) \leq (3 + 6\tilde{\tau})\sqrt{\frac{\lambda'}{n}\left(\frac{1}{2}\log\left(\frac{\lambda'n}{m}\right) + \log\frac{2}{\alpha}\right)}$$

for any $n \geq (91\lambda'(\log(\lambda'n/m) + \log(1/\alpha))) \vee (m/\lambda')$. $\qquad\square$

### E.1 Proofs from Section 4

For convenience, we reproduce the theorem statements.

Define the random variable $\beta_k(n)$ to be the (potentially random) confidence width returned by CSUpdate for estimator $k$ after $n$ updates. For the remainder of the section, we assume that we have access to functions $B_k(n, \delta)$ which have, for all $\delta > 0$,

$$\mathbb{P}(\beta_k(n) \leq B_k(n, \delta)) \geq 1 - \delta.$$

That is, $B_k$ are deterministic upper bounds. We assume that $\beta_k(n)$ is independent of $\beta_j(m)$ for all $j \neq k$. We also recall the definitions that $\Delta_k = c_k \sigma_k^2 - c_{k^*} \sigma_{k^*}^2$ and $a \wedge b = \min\{a, b\}$. Finally, let $\mathcal{C}$ be the event that all the confidence intervals for all estimators and all rounds of the algorithm are correct.

**Theorem 3.** *Assume that the conditions of Theorem 1 hold and that the confidence sequences $u_{k,n}^\eta, u_{k,n}^{(\psi,1)}, u_{k,n}^{(\psi,2)} \to 0$ for all $k \in [K]$ and are all level $\delta/3K$. Then both CS-LUCB and CS-SE with $u_k = (u_k^\eta, u_k^{(\psi,1)}, u_k^{(\psi,2)})$ return an $(\epsilon, \delta)$-PAC index.*

*If we have a deterministic upper bound $B_k(n, \delta)$ such that, for all $\delta > 0$, $\mathbb{P}(\beta_k(n) \leq B_k(n, \delta)) \geq 1 - \delta$, then both algorithms terminate in at most $\sum_{k \in [K]} \min \left\{ n : B_k(n, \delta/K) \leq \frac{\Delta_k}{4} \vee \frac{\epsilon}{2} \right\}$ samples.*

*If, additionally, there exists constants $\nu_\eta$, $\nu_{(\psi,1)}$, and $\nu_{(\psi,2)}$ such that $u_{k,n}^\theta \leq \mathcal{O}(n^{-\nu_\theta} \log(nK/\delta))$ for all $\theta \in \{\eta, (\psi,1), (\psi,2)\}$ and all $k \in [K]$, then the sample complexity is*

$$\mathcal{O}\left( \sum_{k=1}^K (\Delta_k \vee \epsilon)^{-1/\nu} \left( \log \frac{K}{\delta(\Delta_k \vee \epsilon)^{1/\nu}} \right)^{1/\nu} \right),$$

*with probability at least $1 - \delta$, where $\nu = \min\{2\nu_\eta, \nu_{(\psi,1)}, \nu_{(\psi,2)}\}$. In particular, if $\psi(W, \eta)$ is sub-Gaussian, we recover the sample complexity results (up to log factors) of [4, 17] under the mild condition of $\nu_\eta \geq 1/4$.*

*Proof.* Throughout the proof, define let $\mathcal{C}$ be the event of all three confidence sequences being correct; under the assumptions of the theorem, the probability of the three confidence sequences for every estimator being simultaneously correct is at least $1 - K(\alpha/K3 + \alpha/K3 + \alpha/K3) = 1 - \delta$ when we take $\alpha = \delta/3K$. Without loss of generality, we also assume that estimators are sorted in ascending order, that is $c_1 \sigma_1^2 \leq c_2 \sigma^2 \leq \ldots \leq c_K \sigma_K^2$.

**CS-LUCB:** Correctness is a trivial consequence of the termination rule and the confidence sequence guarantees; if the confidence sequences are all correct, then the optimal arm cannot be eliminated. Since the widths go to zero, eventually we will either eliminate all the arms or identify the optimal arm within any arbitrarily small $\epsilon$.

Adapting the original LUCB proof of [17] is difficult in our general setting as we do not assume a specific rate of decay of $B_k(n, \delta)$. Instead, we will follow the simplified analysis from [12].

We will also need to define $N_k(n)$ to be the (random) number of times estimator $k$ was sampled through round $n$ and $\hat{\sigma}_k^2(n)$ to be the variance estimate returned after updating the estimator $n$ times. Define $v = (c_1 \sigma_1^2 + c_2 \sigma_2^2)/2$. Generally, we will expect $c_1 \hat{\sigma}_1^2 - \beta_1(n) \geq v$ and $c_k \hat{\sigma}_k^2 - \beta_k(n) \leq v$ for all other $k$. Let $\mathcal{E}_n$ be the set of estimators at round $n$ that are likely to make an error or have insufficient samples, defined as follows. We include all estimators with confidence bounds that erroneously include $v$ or have not been sampled enough times to have $\beta_k(N_k(n)) \leq \epsilon/2$; precisely, $\mathcal{E}_n$ includes estimator 1 if $c_1 \hat{\sigma}_1^2 + \beta_1(N_1(n)) > v$ or $\beta_1(N_1(n)) \geq \epsilon/2$ and estimator $k \neq 1$ if $c_k \hat{\sigma}_k^2(n) - \beta_k(N_k(n)) < v$ or $\beta_k(N_k(n)) \geq \epsilon/2$.

Let $\mathcal{C}$ be the event where all confidence bounds are correct. By construction and our choice of $\alpha$, $\Pr(\mathcal{C}) \geq 1 - \delta$. By Kalyanakrishnan et al. [17, Lemma 2], we have that if $\mathcal{C}$ holds (the confidence bounds are correct), then in a round $n$ when the algorithm has not terminated, either $l_n$ or $u_n$ must be in $\mathcal{E}_n$ (the original proof carefully checks all the cases); that is,

$$\mathcal{C} \cap \{c_{l_n} \sigma_{l_n}^2 + \beta_{l_n}(n) \geq c_{u_n} \sigma_{u_n}^2 + \beta_{u_n}(n) - \epsilon\} \Rightarrow \{l_n \in \mathcal{E}_n\} \text{ or } \{u_n \in \mathcal{E}_n\}.$$

Hence, under $\mathcal{C}$, we can bound the game length by looking at the number of rounds where $l_n$ or $u_n$ are in $\mathcal{E}_n$.

To this end, define $T_k := \min\{n : B_k(n, \delta/K) \le \frac{\Delta_k}{4} \wedge \frac{\epsilon}{2}\}$, i.e. the minimum number of samples before we can guarantee, with probability $1 - \delta/K$, that $\beta_k(T_k) \le \frac{\Delta_k}{4} \wedge \frac{\epsilon}{2}$. Then, for any $k \ne 1$ and $n \ge T_k$,

$$
\begin{aligned}
c_k \hat{\sigma}_k^2(n) - \beta(n, \delta/K) &\ge c_k \sigma_k^2 - 2\beta(n, \delta/K) \\
&\ge c_k \sigma_k^2 - 2B(n, \delta/K) \\
&\ge v + \frac{c_k \sigma_k^2 - c_1 \sigma_1^2}{2} + \frac{c_k \sigma_k^2 - c_2 \sigma_2^2}{2} - 2B(n, \delta/K) \\
&\ge v + \frac{\Delta_k}{2} - 2B(n, \delta/K).
\end{aligned}
$$

There are two cases; if $\Delta_k \le \epsilon/2$, then $B(n, \delta/K) \le \Delta_k/4$ by construction and

$$
c_k \hat{\sigma}_k^2(n) - \beta(n, \delta/K) \ge v + \frac{\Delta_k}{2} - 2B(n, \delta/K) \ge v.
$$

If $\Delta_k \ge \epsilon/2$, then $B(n, \delta/K) \le \epsilon/2$, leading to

$$
c_k \hat{\sigma}_k^2(n) - \beta(n, \delta/K) \ge v + \frac{\Delta_k}{2} - \epsilon \ge v.
$$

In either case, we have that if $n \ge T_k$, then $k \notin \mathcal{E}_n$ with high probability. An analogous argument can be made for $k = 1$. These statements as a whole show that, with $n \ge T_k$, we will not have any estimators in $\mathcal{E}_n$ with high probability.

We can then conclude that

$$
\begin{aligned}
\sum_n \mathbb{1}\left\{\{l_n, u_n\} \cap \mathcal{E}_n \ne \emptyset\right\} &= \sum_n \sum_{k \in [K]} \mathbb{1}\{u_n = k \text{ or } l_n = k\} \mathbb{1}\{i \in \mathcal{E}_n\} \\
&= \sum_{k \in [K]} \sum_n \mathbb{1}\{u_n = k \text{ or } l_n = k\} \mathbb{1}\{N_k(n) \le T_k\} \le \sum_k T_k,
\end{aligned}
$$

where the last line is because estimator $k$ can only be chosen $T_k$ times before $N_k(n) > T_k$. Adding these terms up, we see that

$$
\sum_{k \in [K]} \min\left\{n : B_k(n, \delta/K) \le \frac{\Delta_k}{4} \vee \frac{\epsilon}{2}\right\},
$$

is a bound on the sample complexity under event $\mathcal{C}$, as claimed.

**Successive Elimination:** We now turn to CS-SE. Recall the definition of $\mathcal{C}$ as the event where the confidence sequeces are correct and that, without loss of generality, we assumed that estimator 1 is optimal.

Recall that the algorithm terminates when either $|S| = 1$ or $\max\{\beta_k : k \in S\} \le \epsilon/2$. Under $\mathcal{C}$, we will argue that, in either case, $S$ contains the optimal arm. Hence, every arm in $S$ is $\epsilon$-sub-optimal.

By way of contradiction, assume that the optimal arm was eliminated before the algorithm terminates. This must have happened on a round with $c_{k^*}(\hat{\sigma}_{k^*}^2 + \beta_{k^*}) \le c_1(\hat{\sigma}_1^2 - \beta_1)$, where $k^*$ corresponds to the minimum $c_k \hat{\sigma}_k^2$ on that round. However, if the confidence bounds are correct, then this inequality would imply that

$$
c_{k^*} \sigma_{k^*}^2 \le c_{k^*}(\hat{\sigma}_{k^*}^2 + \beta_{k^*}) \le c_1(\hat{\sigma}_1^2 - \beta_1) \le c_1 \sigma_1^2,
$$

which is a contradiction. Hence, with probability $1 - \delta$, the best arm is not eliminated.

Now, let $\hat{k}$ be the arm returned by the algorithm. Because the best arm must be in $S$, we have

$$
c_{\hat{k}} \sigma_{\hat{k}}^2 \le \hat{c}_{\hat{k}} \hat{\sigma}_{\hat{k}}^2 + \beta_{\hat{k}} \le c_1 \hat{\sigma}_1^2 + \beta_{\hat{k}} \le c_1 \sigma_1^2 + \beta_1 + \beta_{\hat{k}},
$$

and hence $c_{\hat{k}} \sigma_{\hat{k}}^2 \le c_1 \sigma_1^2 + \epsilon$.

Since we assumed the setting of Theorem 1 and that $u_{k,n}^{\eta}, u_{k,n}^{(\psi,1)}, u_{k,n}^{(\psi,1)} \to 0$, we must have $|\hat{\sigma}_k^2(\mathcal{D}_n) - \sigma_k^2| \to 0$. Thus, there exists an $n_k$ such that $\beta_k \le \epsilon/2$, which it turn implies that

by at most $\sum_k n_k$ samples, each estimator will have received enough samples to either be eliminated or for the algorithm to terminate. Hence, CS-SE is $(\epsilon, \delta)$-PAC.

We now turn to the sample complexity, bounding it separately for every arm. If arm $k$ is eliminated (as opposed to surviving until all $\beta_k \leq \epsilon/2$), it is sufficient that the event $\mathcal{E}_k = \{c_1(\hat\sigma_1^2 + \beta_1) \leq c_k(\hat\sigma_k^2 - \beta_k)\}$ occurs (the arm may be eliminated on a round where $\hat{k} \neq 1$ earlier, so this calculation will be an upper bound). Because

$$c_1(\sigma_1^2 + 2\beta_1) \geq c_1(\hat\sigma_1^2 + \beta_1) \text{ and } c_k(\hat\sigma_k^2 - \beta_k) \geq c_k(\sigma_k^2 - 2\beta_k),$$

$\mathcal{E}_k$ must happen if $c_1(\sigma_1^2 + 2\beta_1) \leq c_k(\sigma_k^2 - 2\beta_k)$, which can be rearranged to

$$\beta_1 + \beta_k \leq \frac{1}{2}(c_k\sigma_k^2 - c_1\sigma_1^2) = \frac{1}{2}\Delta_k.$$

Recall that $\beta_k(n, \alpha)$ was the confidence interval width after $n$ rounds of the algorithm (i.e. $\hat\sigma_k^2$ was updated $n$ times) that holds with probability $\alpha$. Similarly define $S(n)$ to be $S$ after $n$ rounds. The algorithm will terminate if all arms except one are eliminated or if $\beta_k(n, \delta/K) \leq \epsilon/2$ for all $k \in S(n)$. Thus, a sufficient condition for the algorithm's termination is that, for all $k$, either

$$\beta_1(n) + \beta_k(n, \delta/K) \leq \frac{\Delta_k}{2} \text{ or } \beta_k(n) \leq \frac{\epsilon}{2} \forall k \in S(n). \tag{8}$$

One can see that if $\beta_k(n) \leq \frac{\Delta_k}{4} \vee \frac{\epsilon}{2}$ for all $k$, then the condition Eq. (8) is satisfied. Hence, it is sufficient that $B_k(n, \delta/K) \leq \frac{\Delta_k}{4} \vee \frac{\epsilon}{2}$ for all $k$ for a total of

$$\sum_k \min\left\{n : B_k(n) \leq \frac{\Delta_k}{4} \vee \frac{\epsilon}{2}\right\}$$

samples, as claimed.

This sample complexity is a complicated function of the boundary sequences. To establish a connection with the typical bounds in the bandit literature, we will evaluate the complexity for the typical Hoeffding-like tail behavior assuming $u_{k,n}^\theta = \mathcal{O}\left(n^{-\nu_\theta} \log(K/\delta)\right)$ for all $\theta \in \{\eta, (\psi, 1), (\psi, 2)\}$. This implies that $\beta_k = \mathcal{O}\left(n^{-\nu} \log(nK/\delta)\right)$ for $\nu = \min\{2\nu_\eta, \nu_{(\psi,1)}, \nu_{(\psi,2)}\}$. Then, picking $n = \mathcal{O}\left((\Delta_k \vee \epsilon)^{-1/\nu} \left(\log \frac{K}{\delta(\Delta_k \vee \epsilon)^{1/\nu}}\right)^{1/\nu}\right)$ is sufficient because

$$B_k(n, \delta/K) = \mathcal{O}\left(n^{-\nu} \log(nK/\delta)\right)$$

$$= \mathcal{O}\left((\Delta_k \vee \epsilon)^{1/\nu} \left(\log \frac{K}{\delta(\Delta_k \vee \epsilon)}\right)^{-1} \log\left(\frac{K}{\delta}(\Delta_k \vee \epsilon)^{1/\nu} \left(\log \frac{K}{\delta(\Delta_k \vee \epsilon)}\right)^{-1/\nu}\right)\right)$$

$$= \mathcal{O}(\Delta_k \vee \epsilon),$$

where we have ignored log-log terms. We obtain the total sample complexity bound by summing over $k$. $\qquad\square$

## F  Confidence sequences without an UIF

A more explicit version of Theorem 2 is given and proved below.

**Theorem 6.** *Consider an asymptotically linear estimator $\hat\tau_n$ with $L$-Lipschitz influence function $\phi$, and let $\tilde\tau$ be an upper bound on $|\tau|$. For a sequence of datasets $\mathcal{D}_1 \subseteq \mathcal{D}_2 \subseteq \ldots$ with folds $\mathcal{D}_n = \mathcal{D}_n^\eta \cup \mathcal{D}_n^\tau \cup \mathcal{D}_n^\sigma$, and $\mathcal{D}_{n-1}^\eta \subseteq \mathcal{D}_n^\eta$, $\mathcal{D}_{n-1}^\tau \subseteq \mathcal{D}_n^\tau$, and $\mathcal{D}_{n-1}^\sigma \subseteq \mathcal{D}_n^\sigma$. Assume the following boundary sequences hold:*

*1.* $\mathbb{P}\left(\forall n \geq 1 : \left|\mathbb{E}_{\mathcal{D}_n^\sigma}[\phi(W, \eta, \tau)] - \mathbb{E}[\phi(W, \eta, \tau)]\right| \leq u_n^{(\phi,1)}\right) \geq 1 - \alpha,$

*2.* $\mathbb{P}\left(\forall n \geq 1 : \left|\mathbb{E}_{\mathcal{D}_n^\sigma}[\phi(W, \eta, \tau)^2] - \mathbb{E}[\phi(W, \eta, \tau)^2]\right| \leq u_n^{(\phi,2)}\right) \geq 1 - \alpha,$

*3.* $\mathbb{P}\left(\forall n \geq 1 : \|\hat\eta(\mathcal{D}_n^\eta) - \eta\| \leq u_n^\eta\right) \geq 1 - \alpha$, *and*

4. $\mathbb{P}\left(\forall n \geq 1 : |\hat{\tau}_n(\mathcal{D}_n^\tau, \hat{\eta}(\mathcal{D}_n^\eta)) - \tau| \leq u_n^\tau\right) \geq 1 - \alpha.$

*Then, for the estimator defined by Eq. (1),*

$$\mathbb{P}\left(\forall n \geq 1 : \left|\hat{\sigma}^2(\mathcal{D}_n) - \sigma^2\right| \leq L^2(u_n^\eta + u_n^\tau)^2 + u_n^{(\phi,2)} + \left(u_n^{(\phi,1)}\right)^2 + 2\tilde{\tau}u_n^{(\phi,1)}\right) \geq 1 - 4\alpha.$$

*Proof.* We aim to decompose $|\hat{\sigma}^2(\mathcal{D}_n) - \sigma^2|$ into the terms with confidence sequences, then apply the confidence sequences and a union bound. Throughout the proof we use the shorthand $\hat{\eta}_n := \hat{\eta}(\mathcal{D}_n^\eta)$ and $\hat{\tau}_n := \hat{\tau}(\mathcal{D}_n^\tau, \hat{\eta}_n)$. Also, let $\mathbb{E}_n$ denote the empirical expectation w.r.t. $\mathcal{D}_n^\sigma$.

We can reuse the analysis framework of the UIF case by defining $\psi(W, \eta, \tau) := \phi(W, \eta, \tau) + \tau$ so that $\mathbb{E}[\psi(W, \eta, \tau)] = \tau$ and $\text{var}(\psi(W, \eta, \tau) = \mathbb{E}[\phi(W, \eta, \tau)^2]$; in the UIF case, $\psi$ was not a function of $\tau$. Using the identity $\text{var}_n[\psi(W, \hat{\eta}_n, \hat{\tau}_n)] = \mathbb{E}_n[\psi(W, \hat{\eta}_n, \hat{\tau}_n)^2] - \mathbb{E}_n[\psi(W, \hat{\eta}_n, \hat{\tau}_n)]^2$, we can expand $\hat{\sigma}^2(\mathcal{D}_n) - \sigma^2$ as

$$
\begin{aligned}
\hat{\sigma}^2(\mathcal{D}_n) - \sigma^2 &= \text{var}_n[\psi(W, \hat{\eta}_n, \hat{\tau}_n)] - \text{var}[\psi(W, \eta, \tau)] \\
&= \mathbb{E}_n[\psi(W, \hat{\eta}_n, \hat{\tau}_n)^2] - \mathbb{E}[\psi(W, \eta, \tau)^2] - \mathbb{E}_n[\psi(W, \hat{\eta}_n, \hat{\tau}_n)]^2 + \mathbb{E}[\psi(W, \eta, \tau)]^2 \\
&= \mathbb{E}_n[\psi(W, \hat{\eta}_n, \hat{\tau}_n)^2] - \mathbb{E}_n[\psi(W, \eta, \tau)^2] - \mathbb{E}_n[\psi(W, \hat{\eta}_n, \hat{\tau}_n)]^2 + \mathbb{E}_n[\psi(W, \eta, \tau)]^2 \\
&\quad + \left(\mathbb{E}_n[\psi^2(W, \eta, \tau)] - \mathbb{E}[\psi(W, \eta, \tau)^2]\right) + \left(\mathbb{E}_n[\psi(W, \eta, \tau)]^2 - \mathbb{E}[\psi(W, \eta, \tau)]^2\right) \\
&= \mathbb{E}_n\left[(\psi(W, \hat{\eta}_n, \hat{\tau}_n) - \psi(W, \eta, \tau))^2\right] + 2\mathbb{E}_n\left[\psi(W, \eta, \tau)(\psi(W, \hat{\eta}_n, \hat{\tau}_n) - \psi(W, \eta, \tau))\right] \\
&\quad - \mathbb{E}_n[\psi(W, \hat{\eta}_n, \hat{\tau}_n) - \psi(W, \eta, \tau)]\mathbb{E}_n[\psi(W, \hat{\eta}_n, \hat{\tau}_n) + \psi(W, \eta, \tau)] \\
&\quad + \left(\mathbb{E}_n[\psi(W, \eta, \tau)^2] - \mathbb{E}[\psi(W, \eta, \tau)^2]\right) + \left(\mathbb{E}_n[\psi(W, \eta, \tau)]^2 - \mathbb{E}[\psi(W, \eta, \tau)]^2\right).
\end{aligned}
$$

Rearranging the second and third terms and using Cauchy-Schwarz, we have

$$
\begin{aligned}
&\left|2\mathbb{E}_n\left[\psi(W, \eta, \tau)(\psi(W, \hat{\eta}_n, \hat{\tau}_n) - \psi(W, \eta, \tau))\right] - \mathbb{E}_n[\psi(W, \hat{\eta}_n, \hat{\tau}_n) - \psi(W, \eta, \tau)]\mathbb{E}_n[\psi(W, \hat{\eta}_n, \hat{\tau}_n) + \psi(W, \eta, \tau)]\right| \\
&= \left|\mathbb{E}_n\left[(2\psi(W, \eta, \tau) - \mathbb{E}_n[\psi(W, \hat{\eta}_n, \hat{\tau}_n) + \psi(W, \eta, \tau)])(\psi(W, \hat{\eta}_n, \hat{\tau}_n) - \psi(W, \eta, \tau))\right]\right| \\
&\leq \sqrt{\mathbb{E}_n\left[(2\psi(W, \eta, \tau) - \mathbb{E}_n[\psi(W, \hat{\eta}_n, \hat{\tau}_n) + \psi(W, \eta, \tau)])^2\right]}\sqrt{\mathbb{E}_n\left[(\psi(W, \hat{\eta}_n, \hat{\tau}_n) - \psi(W, \eta, \tau))^2\right]} \\
&= \sqrt{\mathbb{E}_n\left[(\psi(W, \eta, \tau) - \psi(W, \hat{\eta}_n, \hat{\tau}_n))^2\right]}\sqrt{\mathbb{E}_n\left[(\psi(W, \hat{\eta}_n, \hat{\tau}_n) - \psi(W, \eta, \tau))^2\right]} \\
&= \mathbb{E}_n\left[(\psi(W, \hat{\eta}_n, \hat{\tau}_n) - \psi(W, \eta, \tau))^2\right].
\end{aligned}
$$

With the triangle inequality, we have

$$
\begin{aligned}
|\hat{\sigma}^2(\mathcal{D}_n) - \sigma^2| &\leq 2\mathbb{E}_n\left[(\psi(W, \hat{\eta}_n, \hat{\tau}_n) - \psi(W, \eta, \tau))^2\right] + \left|\mathbb{E}_n[\psi(W, \eta, \tau)^2] - \mathbb{E}[\psi(W, \eta, \tau)^2]\right| \\
&\quad + \left|\mathbb{E}_n[\psi(W, \eta, \tau)]^2 - \mathbb{E}[\psi(W, \eta, \tau)]^2\right|.
\end{aligned}
$$

For the first term, we use the Lipschitz property of $\psi$, providing

$$\left|\mathbb{E}_n\left[(\psi(W, \hat{\eta}_n, \hat{\tau}_n) - \psi(W, \eta, \tau))^2\right]\right| \leq L^2(\|\hat{\eta}_n - \eta\| + |\hat{\tau}_n - \tau|)^2. \tag{9}$$

The second term is easily bounded by the assumptions, since

$$\left|\mathbb{E}_n[\psi^2(W, \eta, \tau)] - \mathbb{E}[\psi(W, \eta, \tau)^2]\right| = \left|\mathbb{E}_n[\phi^2(W, \eta, \tau)] - \mathbb{E}[\phi(W, \eta, \tau)^2]\right| \leq u_n^{(\phi,2)}.$$

We can tackle the third term by using a similar expansion. Recalling that $\mathbb{E}[\psi(W, \eta, \tau)] = \tau$,

$$
\begin{aligned}
&\left|\mathbb{E}_n[\psi(W, \eta, \tau)]^2 - \mathbb{E}[\psi(W, \eta, \tau)]^2\right| \\
&\leq \left|(\mathbb{E}_n[\psi(W, \eta, \tau)] - \mathbb{E}[\psi(W, \eta, \tau)])^2 + 2\mathbb{E}[\psi(W, \eta, \tau)](\mathbb{E}_n[\psi(W, \eta, \tau)] - \mathbb{E}[\psi(W, \eta, \tau)])\right| \\
&= \left|(\mathbb{E}_n[\phi(W, \eta, \tau)] - \mathbb{E}[\phi(W, \eta, \tau)])^2 + 2\tau(\mathbb{E}_n[\psi(W, \eta, \tau)] - \mathbb{E}[\psi(W, \eta, \tau)])\right| \\
&\leq \left(u_n^{(\phi,1)}\right)^2 + 2\tilde{\tau}u_n^{(\phi,1)}.
\end{aligned}
$$

Collecting the terms above will yield the theorem statement. $\square$

While the theorem in stated explicitly in terms of the convergence rate of $|\hat{\tau}_n - \tau|$, bounding this term by quantities depending on $\|\hat{\eta}_n - \eta\|^2$ is the study of much of the double machine learning literature. In particular, one can notice that $\hat{\tau}_n$ is fit via Empirical Risk Minimization, as long as one associates the influence function with a loss function, the results from Foster and Syrgkanis [5, Section 4] provide explicit convergence rates provided that $\phi$ is convex in $\tau$ and the complexity of the nuisance function class is bounded. Taken as a whole, the width of the confidence sequence scales with $\mathcal{O}(\|\hat{\eta}(\mathcal{D}_n^\eta) - \eta\|^2)$.