# OpenReview forum: "Asymptotically Best Causal Effect Identification with Multi-Armed Bandits"
_NeurIPS.cc/2021/Conference — NeurIPS 2021 Poster_

### Official Review · Reviewer_sw9W · 2021-06-25

**Rating:** 3
**Confidence:** 4

**Summary:**

This paper proposed two bandit methods (CS-LUCB and CS-SE) that learn the best causal identification formula in terms of the cost-adjusted asymptotic variance. In the two bandit algorithms, they replaced the usual empirical reward terms with the empirical variance estimations. They showed their methods find the best formula under some conditions and also provided sample complexities.

**Limitations And Societal Impact:**

I don't see any potential negative societal impact.

**Main Review:**

Originality:
Using bandit framework to learn the best causal identification terms is new to me. The way they mixed the variance estimation terms in standard algorithms such as LUCB and SE was trivial.

Quality:
In my opinion, this paper was written very poorly.
First, the notations and terms were defined very unclearly. For example, in line 76, what are \lambda_x, what do you mean by causal contrast? in the paragraph starting in line 97, what is the function \phi? why asymptotic linear? etc.
Second, several (in)equalities are incorrect. E.g. in line 81, the decomposition is not correct, there should be a do operator over X=x in the condition of \mu_x(Z_1). Line 82 has similar issue as well, the decomposition does not hold. If interventional quantities, i.e. do operators need to exist in the decompositions, it will contradict with the authors' claim that they express the causal contrast using only observational quantities. This example doesn't make sense to me.  In the inequality below line 132, the event in the probability bracket is not even a random event.
Third, theorems hold under too many conditions. It's hard to interpret their sample complexity results theoretically.
Also, there is no practical motivation for the problem.
Typos: 'casual' -> 'causal' in the title.

Clarity:
The paper was poorly written.

Significance:
Learning the causal expression adaptively is a worth-pursuing problem, but this paper is not good enough.

**Time Spent Reviewing:**

2

---

> ### Author Response · Authors · 2021-08-12
> **Thanks for the review!**
>
> We regret that you found the presentation confusing. We provide specific answers to your questions below and we will work on improving the exposition.
>
> __Clarifications__
>
> The formulas at lines 81 and 82 are correct. These formulas are based on the adjustment (or backdoor) criterion and on the frontdoor criterion respectively, which are very well established in the causal inference literature. These criteria can be found e.g. in the online Chapter 3 of the book ‘Causality’ by Judea Pearl (http://bayes.cs.ucla.edu/BOOK-2K/ch3-3.pdf), equation (3.21) at page 125 and equation (3.31) at page 129 respectively.
> For example, the frontdoor criterion in equation (3.31) states (with slightly different notation) that $p(Y|do(X=x))=\sum_z p(Z=z|X=x) \sum_{x’}p(X=x’)p(Y|X=x’,Z=z)$. We take the expectation with respect to Y, obtaining $\sum_y y p(Y=y|do(X=x))=\sum_z p(Z=z|X=x) \sum_{x’} p(X=x’) \sum_y y p(Y=y|X=x’,Z=z)$.
>
> $\lambda_x$ indicates a scalar value that depends on the value of x. For example, for $X$ taking value 1 and 0 and $\lambda_1=1$ and $\lambda_0=-1$, we obtain the Average Treatment Effect (ATE) as causal contrast. The definition of $\lambda_x$ was mistakenly removed, we apologize for that. The term causal contrast is commonly used in the causal inference literature to describe a measure of causal effect such as e.g. the ATE.
> If we define $k^*$ as the arm returned by the algorithm (as described by the experimental protocol), then at least the event is random. We agree that this notational choice is pretty unclear and we will change it.
>
> __Conditions of the Theorem__
>
> We disagree that the theorems require too many conditions; if any of the conditions are dropped, then no confidence intervals can exist. Theorem 1 is written in generality to emphasize the secord-order scaling with the nuisance estimate, but Corollary 1 gives a specific instance of the theorem under the assumption of sub-Gaussian noise, the most common assumption in the bandit literature.
>
> __Sample Complexity Results__
>
> The sample complexity results follow the same form in the rest of the best-arm-identification literature, so at least the lack of interpretability is an inherited problem. Importantly, these sample complexity bounds match (to leading order) lower bounds for the problem, which we can transport to our setting if we make strong assumptions about the decay rates of the confidence sequences. However, as discussed with other reviewers, lower bounds are, in general, much messier in our setting since any instance dependent bound will have to involve not only the gaps but also how quickly the confidence sequences shrink.
>
> __Motivation__
>
>  We agree that we did not sufficiently motivate our contribution. Our motivation for this work can be summarized as follows.
> First, we wanted to overcome the three following current limitations on the selection of identification formulas based on asymptotic variance:
> - Inability to select the best identification formula in some latent variables cases. As a concrete example, the DAG $Z_1\rightarrow Z_2 \leftarrow Z_3\rightarrow Y$, $Z_1\rightarrow A\rightarrow Y$ admits the following adjustment sets: $\emptyset$, $\\{Z_1\\}$, $\\{Z_3\\}$, $\\{Z_1, Z_2\\}$,$\\{Z_1, Z_3\\}$, $\\{Z_2, Z_3\\}$, and $\\{Z_1, Z_2, Z_3\\}$. According to Theorem 5 in the Appendix, the set $\\{Z_3\\}$ has the lowest asymptotic variance. If, however, $Z_3$ is latent, then either $\\{Z_1, Z_2\\}$ or the empty set has the lowest variance depending on the model distribution, and therefore graphical criteria are unable to return the adjustment set with smallest variance. Our algorithm would be able to resolve this case.
> - Applicability to adjustment criteria only.
> - Inability to account for the costs of observing covariates.
>
> In addition, we wanted to allow for an adaptive covariate measurement (ACM) setting, namely
> - A setting where one can choose which covariates to measure adaptively and wishes to identify the best formula as efficiently as possible.
>
> Current causal inference literature does not consider this setting despite it being relevant to many applications. Consider, for example, a large clinical trial in which we have the option of using several different estimators. Before the trial begins, we wish to identify, using a group of individuals, the estimator with the best long-run behavior, i.e. with the lowest cost-adjusted asymptotic variance. One approach would be to assign an estimator to each individual randomly, collect the covariates required by the estimator, and then estimate the estimator’s asymptotic variance. However, using a bandit algorithm to assign an estimator to the $i$th individual based on the feedback from the individuals $k$ for $j<i$ would have the advantage of requiring fewer individuals. Our idea of framing the ACM setting as a best-arm-identification bandit problem is
>
> __Technical Contributions__
>
> While the paper proposes few innovations to bandit algorithms, in our opinion Theorem 1 represents a solid technical contribution. We realized that we did not sufficiently highlight and explain its significance in the submission. We attempt to do this below (we will also incorporate the explanation in the paper). Theorem 1 implies that having an uncentered influence function is a sufficient condition for obtaining the second-order dependence on the rate of nuisance function estimation. Neyman orthogonality, while sufficient for predicting the causal contrast at a parametric rate, is not sufficient for predicting the variance at the parametric rate. The proof of Theorem 1 is straightforward under the condition of having an uncentered influence function, but a second-order result is generally not possible without this condition; most of the challenge was in identifying it. Another key observation made during the proof of Theorem 1 was that the error admits the following two decompositions:
> A decomposition that would require to control $\sum_{i=1}^n \psi(w_i, \hat\eta_n)$ for all sequences of estimators $\hat\eta_n$ of $\eta$.
>  A decomposition that would only require a confidence sequence for $\sum_{i=1}^n \psi(w_i, \eta)$, i.e. at the true $\eta$, albeit with a few more details in the analysis. Using the second decomposition allowed us to generalize all our results from confidence intervals to confidence sequences because we only needed to control the deviations at $\eta$.

---

### Official Review · Reviewer_bePw · 2021-07-16

**Rating:** 4
**Confidence:** 3

**Summary:**

The author considers the problem of selecting the best (in terms of variance) estimator among K estimators in an online manner. The estimators class is limited to the ones that satisfy the asymptotic linearity and its asymptotic variance can be computed from the uncentered influence function. The author considered this problem setting as a best-arm identification problem and applied the SE and LUCB algorithms. The variance computation is based on the sample splitting approach. The authors conducted theoretical analyses of its performance as well as numerical experiments on synthetic data.

**Ethical Concerns:**

None.

**Limitations And Societal Impact:**

I don't think there is a problem.

**Main Review:**

I do not see the novelty of this problem setting. It seems to me that this is just an application of BAI with the arm’s mean as the value of the variance. The computation of the confidence bound for the variance does not look novel. The proof of Theorem 1 looks straightforward to me. (please point out the difficulties of the proof) If the sample splitting idea is based on Chernozhukov et al 2018, the authors should cite it in the main paper and proofs. Even if it is a different problem setting, it would be strange not to cite the causal bandit papers (e.g., [1], [2]). The differences from those studies should be compared and carefully discussed. In some places, (even in the title) there are typos "casual". No sample complexity lower bounds are given. The title having "Best" causal effect identification, it is a bit misleading to put the word "Best" in the title, since the authors are not showing the lower bound and a method that matches the lower bound. LUCB and SE are not the state of the art algorithms for the fixed confidence BAI (see e.g., [3], [4]).

", we could consider the problem of trying to obtain the best estimate of the causal effect given a limited budget. This problem is more analogous to the cumulative regret minimizing bandit problem"

I don't think this problem resembles the expected cumulative regret minimization problem. Might be similar to the problem of fixed budget BAI.




[1] Lattimore et al, Causal Bandits: Learning Good Interventions via
Causal Inference, NeurIPS 2016

[2] Kallus, Instrument-Armed Bandits, ALT 2018

[3] Non-Asymptotic Pure Exploration by Solving Games, Neurips 2019.

[4] Garivier and Kaufmann, Optimal Best Arm Identification with Fixed Confidence, COLT 2016.


---------- After Rebuttal --------

After reading your rebuttals and other reviewers' comments, my score remains the same.
Technical contributions can be made much clearer and I think the work needs a clear comparison with the existing pure exploration literature.
The argument on cumulative regret still does not make sense to me. You need to prove the equivalence to claim the similarity I think.
Furthermore, again the typo "casual" is in your rebuttal. Please make sure to proofread.




**Time Spent Reviewing:**

5

---

> ### Author Response · Authors · 2021-08-12
> **Thanks for the review!**
>
> Thanks for your review. Hopefully, our explanation below will resolve some of your concerns about the novelty and technical contribution of our work.
>
> __Novelty__
>
> The goal of this work was to contribute to the field of causal inference, rather than to the field of bandits.
> We believe the contributions to causal inference are significant. The problem of selection of identification formulas based on asymptotic variance has only started to receive some attention and is still at its infancy. Before the recently established graphical criteria, the set of parents or the smallest set of covariates was often used as adjustment sets, which is often inefficient. Whilst representing a huge advancement, graphical criteria have some limitations.
>
> As also explained to the other reviewer, we wanted to overcome the following three current limitations on selection of identification formulas based on asymptotic variance:
> Inability to select the best identification formula in some latent variables cases. As a concrete example, the DAG $Z_1\\rightarrow Z_2 \\leftarrow Z_3\rightarrow Y$, $Z_1\rightarrow A\rightarrow Y$ admits the following adjustment sets: $\\emptyset$, $\\{Z_1\\}$, $\\{Z_3\\}$, $\\{Z_1, Z_2\\}$,$\\{Z_1, Z_3\\}$, $\\{Z_2, Z_3\\}$, and $\\{Z_1, Z_2, Z_3\\}$. According to Theorem 5 in the Appendix, the set $\{Z_3\}$ has the lowest asymptotic variance. If, however, $Z_3$ is latent, then either $\\{Z_1, Z_2\\}$ or the empty set has the lowest variance depending on the model distribution, and therefore graphical criteria are unable to return the adjustment set with smallest variance. Our algorithm would be able to resolve this case.
> Applicability to adjustment criteria only.
> Inability to account for the costs of observing covariates.
> In addition, we wanted to allow for an adaptive covariate measurement (ACM) setting, namely
> A setting where one can choose which covariates to measure adaptively and wishes to identify the best formula as efficiently as possible.
>
> Current causal inference literature does not consider this setting despite it being relevant to many applications. Consider, for example, a large clinical trial in which we have the option of using several different estimators. Before the trial begins, we wish to identify, using a group of individuals, the estimator with the best long-run behavior, i.e. with the lowest cost-adjusted asymptotic variance. One approach would be to assign an estimator to each individual randomly, collect the covariates required by the estimator, and then estimate the estimator’s asymptotic variance. However, using a bandit algorithm to assign an estimator to the $i$th individual based on the feedback from the individuals $k$ for $j<i$ would have the advantage of requiring fewer individuals. Our idea of framing the ACM setting as a best-arm-identification bandit problem is simple, yet effective. We also believe that our work may serve as a first step for solving more general problems.
>
> We agree that the contribution to the bandit literature is minimal, but as explained above, contributing to this field was not our goal. The analysis for the bandit algorithms we use are quite standard; both algorithms we used are trivial to extend to the confidence sequence case. However, state-of-the-art algorithms like lil’UCB, Exponential Gap Elimination, and stop-and-track, are not easily adapted to use confidence sequences that have widths which decrease at arbitrary rates (more details are provided below.
>
> __Technical Contributions__
>
> While the paper proposes few innovations to bandit algorithms, in our opinion Theorem 1 represents a solid technical contribution. We realized that we did not sufficiently highlight and explain its significance in the submission. We attempt to do this below (we will also incorporate the explanation in the paper). Theorem 1 implies that having an uncentered influence function is a sufficient condition for obtaining the second-order dependence on the rate of nuisance function estimation. Neyman orthogonality, while sufficient for predicting the causal contrast at a parametric rate, is not sufficient for predicting the variance at the parametric rate. The proof of Theorem 1 is straightforward under the condition of having an uncentered influence function, but a second-order result is generally not possible without this condition; most of the challenge was in identifying it. Another key observation made during the proof of Theorem 1 was that the error admits the following two decompositions:
> A decomposition that would require to control $\sum_{i=1}^n \psi(w_i, \hat\eta_n)$ for all sequences of estimators $\hat\eta_n$ of $\eta$.
>  A decomposition that would only require a confidence sequence for $\sum_{i=1}^n \psi(w_i, \eta)$, i.e. at the true $\eta$, albeit with a few more details in the analysis. Using the second decomposition allowed us to generalize all our results from confidence intervals to confidence sequences because we only needed to control the deviations at $\eta$.
>
> __Comparison to Causal Bandits__
>
>
> As related work, we discuss the part of the causal inference literature that is concerned with the selection of formulas expressing a causal quantity as a function of observational data. We felt that, as casual bandits have a radically different goal, namely to estimate a causal quantity from interventional data, discussing them was not that relevant, but we are happy to add a general discussion on them.
>
> Lower bounds are not easy to formulate for general confidence sequences, as the sample complexities depend on how quickly the widths of the confidence sequence decrease as well as the gaps. For example, the BAI problem with general confidence sequences does not easily allow for implementations of the optimal fixed-confidence BAI algorithms. Track-and-stop style algorithms rely on parametric assumptions about the reward distributions while all we can guarantee is a frequentist confidence sequence on the reward. Minimax-optimal algorithms, such as lil’UCB and Exponential Gap Elimination, cannot be easily transported either for the following reasons. The stopping condition of lil’UCB is triggered when one arm is sampled at least a constant proportion, but the proof of correctness requires that the confidence bound widths shrink at the same rate for every arm, which is not necessarily true in our framework. The same difficulty arises in Exponential Gap Elimination and even Median Elimination; both algorithms also assume a uniform rate of delay of the bound widths across arms. Hence, we were forced to use the simpler algorithms LUCB and SE, which allow us to derive correctness and sample complexity results when the confidence intervals decay at arbitrary speeds. The difficulty of an instance will need to be a function of the shape of the entire confidence sequence, so it will only be easy to adapt the lower bounds to the case where all the confidence sequences are decaying at the same rate.
>
> __“Best” Arm Identification__
>
> We meant “asymptotically best” to indicate which estimator we are searching for, analogous to how “best-arm-identification” finds the best arm. Perhaps “finding the asymptotically best causal effect estimator” would be less ambiguous?
>
> __Cumulative Regret__
>
> We were thinking about the setting where the goal is to directly obtain an estimate of $\tau$ that is as tight as possible, where the regret is defined w.r.t. to the comparator that allocated all the samples to the optimal estimator. For each round, the learner chooses an estimator to sample, updates that estimator, and updates the best estimate of $\tau$ which uses information from all estimators. In this sense, the reward of each action is how much the uncertainty was reduced. This problem is substantially harder, as it requires general methods for combining estimates from different estimators with finite-sample confidence bounds. However, we agree that “we could consider the problem of trying to obtain the best estimate of the causal contrast given a limited budget. This problem is more analogous to the cumulative regret minimizing bandit problem” is unclear and we will improve the discussion.

---

### Official Review · Reviewer_ysqL · 2021-07-16

**Rating:** 5
**Confidence:** 1

**Summary:**

This paper uses the best-arm identification framework to select an appropriate estimator for identifying a causal relationship between two variables. Such an estimator takes the observed data as input to quantify the causal effect between the variables of interest. In the setup proposed in this paper, each arm corresponds to an estimator, and the goal of the learner is to select the best estimator (the one with lowest variance), while also considering the cost associated with the estimator (available apriori).

The primary contribution of this paper is a confidence bound for the variance of such estimators under assumptions such as Lipschitz continuity on the “uncentered influence function”. This bound is then used in two standard best-arm identification algorithms (a UCB based strategy and an elimination algorithm) to select the “best” arm. The paper derives upper bounds on the sample complexity of both the algorithms. Experiments show that the algorithms perform favorably as compared to a random strategy.

**Limitations And Societal Impact:**

The authors discuss a limitation in L364-365, however, even before that they should highlight the limitations of posing this problem as a best-arm identification problem.

**Main Review:**

Originality – Posing the problem of selecting the best estimator for causal effect identification as a best-arm identification problem appears to be new. The confidence bound and the best-arm identification algorithms (and their analysis) are fairly standard.

Quality – I have the following issues:
1.	The paper lacks a clear motivation and does not provide a satisfactory answer as to why it is a good idea to formulate this problem as a best-arm identification problem. Are there any limitations of this formulation?
2.	It is hard for me to assess why the technical contributions in this paper are significant enough. I would be happy to revise my score if the authors can provide simple explanations for this.
3.	Algorithm 1 does not take cost into account. Moreover, shouldn’t the condition inside the first if predicate be: \hat{\sigma}^2_{l_t} + \beta_{l_t} \leq \hat{\sigma}^2_{u_t} - \beta_{u_t} + \epsilon?
4.	L372 is confusing. I don’t understand why regret minimization will be better objective in this case if the goal is to find the best estimator?

Clarity –
1.	It was hard for me to follow the paper. It would be better to clarity in the beginning itself as to what is being done adaptively. L60 gives the impression that the goal is to adaptively select observations to reduce the variance of an estimator. However, the actual goal appears to be slightly different. Once an estimator (an arm) is chosen, collecting the samples for this estimator is not done adaptively. The goal instead is to decide whether it is reasonable to collect more data for an estimator before ruling it out, not to decide what data has to be sampled for a particular estimator. Am I missing something?
2.	In L76, lambda was used without any explanation.
3.	More details in the paragraph that starts at L108 will be helpful. In particular, taking a concrete example of \eta and \psi here will make things easier to follow.

Significance – Please see the 2nd point under quality.

Strengths – The bandit formulation of the problem seems to be new.

Weaknesses – The paper is difficulty to follow (see more details above)

**Time Spent Reviewing:**

15

---

> ### Author Response · Authors · 2021-08-12
> **Thanks for the review!**
>
> Thanks for your helpful feedback! Despite a confidence score of 1, we feel that you understood the paper well.
>
> __Originality__
>
> While the bandit algorithms and the sample complexity results are mostly standard, we believe that the confidence bound is novel and a  solid contribution in its own right. Please see the _Technical Contributions_ section below.
>
> __Quality__
>
> We will respond to each point below.
>
> _Motivation_:
>  We agree that we did not sufficiently motivate our contribution. Our motivation for this work can be summarized as follows.
> First, we wanted to overcome the three following current limitations on the selection of identification formulas based on asymptotic variance:
> - Inability to select the best identification formula in some latent variables cases. As a concrete example, the DAG $Z_1\rightarrow Z_2 \leftarrow Z_3\rightarrow Y$, $Z_1\rightarrow A\rightarrow Y$ admits the following adjustment sets: $\emptyset$, $\\{Z_1\\}$, $\\{Z_3\\}$, $\\{Z_1, Z_2\\}$,$\\{Z_1, Z_3\\}$, $\\{Z_2, Z_3\\}$, and $\\{Z_1, Z_2, Z_3\\}$. According to Theorem 5 in the Appendix, the set $\\{Z_3\\}$ has the lowest asymptotic variance. If, however, $Z_3$ is latent, then either $\\{Z_1, Z_2\\}$ or the empty set has the lowest variance depending on the model distribution, and therefore graphical criteria are unable to return the adjustment set with smallest variance. Our algorithm would be able to resolve this case.
> - Applicability to adjustment criteria only.
> - Inability to account for the costs of observing covariates.
>
> In addition, we wanted to allow for an adaptive covariate measurement (ACM) setting, namely
> - A setting where one can choose which covariates to measure adaptively and wishes to identify the best formula as efficiently as possible.
>
> Current causal inference literature does not consider this setting despite it being relevant to many applications. Consider, for example, a large clinical trial in which we have the option of using several different estimators. Before the trial begins, we wish to identify, using a group of individuals, the estimator with the best long-run behavior, i.e. with the lowest cost-adjusted asymptotic variance. One approach would be to assign an estimator to each individual randomly, collect the covariates required by the estimator, and then estimate the estimator’s asymptotic variance. However, using a bandit algorithm to assign an estimator to the $i$th individual based on the feedback from the individuals $k$ for $j<i$ would have the advantage of requiring fewer individuals. Our idea of framing the ACM setting as a best-arm-identification bandit problem is simple, yet effective. We also believe that our work may serve as a first step for solving more general problems.
>
>
> _Limitations_: Perhaps the greatest limitation of the best-arm-identification formulation is that it only makes sense when the sample complexity needed to learn the best estimator is smaller than the sample complexity needed to estimate $\tau$ to the desired precision. Other limitations are:
> - Using different data in the estimate of multiple asymptotic variances. Reusing data between estimators would clearly be more sample efficient, but the error would become dependent and the analysis would be difficult.
> - Lack of  finite-sample guarantees on the estimate of $\tau$.
>
> We will add these limitations in the Discussion section.
>
> _Technical Contributions_:
> The analysis in bandits and the bandit algorithms we use are quite standard. However, the goal of this paper was not to achieve an algorithmic advance in bandits, but rather to provide a solution to open problems in causal inference.
> Both bandit algorithms are trivial to extend to the confidence sequence case. However, in our opinion, Theorem 1 represents a solid technical contribution. We realized that we did not sufficiently highlight and explain its significance in the submission. We attempt to do this below (we will incorporate the explanation also in the paper). Theorem 1 implies that having an uncentered influence function is a sufficient condition for obtaining the second-order dependence on the rate of nuisance function estimation. Neyman orthogonality, while sufficient for predicting the causal contrast at a parametric rate, is not sufficient for predicting the variance at the parametric rate. The proof of Theorem 1 is straightforward under the condition of having an uncentered influence function, but a second-order result is generally not possible without this condition; most of the challenge was in identifying it. Another key observation made during the proof of Theorem 1 was that the error admits the following two decompositions:
> 1. A decomposition that would require to control $\sum_{i=1}^n \psi(w_i, \hat\eta_n)$ for all sequences of estimators $\hat\eta_n$ of $\eta$.
> 2. A decomposition that would only require a confidence sequence for $\sum_{i=1}^n \psi(w_i, \eta)$, i.e. at the true $\eta$, albeit with a few more details in the analysis. Using the second decomposition allowed us to generalize all our results from confidence intervals to confidence sequences because we only needed to control the deviations at $\eta$.
>
> _Algorithm 1_:  We apologize for the typos: all $\hat\sigma_k^2$ (or upper and lower bounds)) should be scaled by the costs, and the correct predicate should be $c_{l_t}\left(\hat\sigma_{l_t}^2 + \beta_{l_t}\right) \leq c_{u_t}\left(\hat\sigma_{u_t}^2 - \beta_{u_t}\right) - \epsilon$.
>
> _Cumulative Regret_: We were thinking about the setting where the goal is to directly obtain an estimate of $\tau$ that is as tight as possible, where the regret is defined w.r.t. to the comparator that allocated all the samples to the optimal estimator. For each round, the learner chooses an estimator to sample, updates that estimator, and updates the best estimate of $\tau$ which uses information from all estimators. In this sense, the reward of each action is how much the uncertainty was reduced. This problem is substantially harder, as it requires general methods for combining estimates from different estimators with finite-sample confidence bounds. We will clarify these statements in the paper.
>
>
>
> __Clarity__
>
> _Hard to follow_: Thanks for sharing this point. We will improve the explanation of the Experimental Protocol box and of the bottom of page 3 and move them earlier in the text. You correctly inferred the setting and are not missing anything, thanks for your effort in understanding the setting despite a lack of clarity on our side.
>
> _Defining $\lambda$_: $\lambda_x$ indicates a scalar value that depends on the value of $X$. For example, for $X$ taking values $1$ and $0$, and for $\lambda_1=1$ and $\lambda_0=-1$, we obtain the Average Treatment Effect (ATE) as causal contrast. The definition of $\lambda_x$ was mistakenly removed, we apologize for that.
>
> _Variance estimator_: A previous version of the paper described in detail the estimator of $\sigma^2$ for the influence function of the AIPW, but this paragraph was removed for space. We will prioritize putting it back in.

---

### Official Review · Reviewer_t43v · 2021-07-20

**Rating:** 5
**Confidence:** 3

**Summary:**

This paper studies the problem of identifying the formula with the best asymptotic variance among a set of formulas used to estimate a causal effect, and frames it as a best-arm identification (BAI) problem. The paper develops finite-sample confidence sequences with nuisance functions for the proposed sample-splitting estimator of the asymptotic variance. With these confidence sequences, the paper can modify the existing algorithms LUCB and successive elimination for the studied BAI problem and prove sample complexity bounds in terms of the estimation rate of the nuisance function.

**Ethical Concerns:**

It seems that there is no ethical concern.

**Limitations And Societal Impact:**

The paper briefly mentions the limitations in the Discussion section and it seems there is no potential negative societal impact. Please address my comments.

**Main Review:**

The paper is well organized and relatively clearly written. The main technical result is developing finite-sample confidence sequences with nuisance functions for the proposed sample-splitting estimator of the asymptotic variance. With this, it is not hard to modify the existing LUCB and successive elimination algorithms for the studied BAI problem and to use the standard techniques in BAI literature to show the corresponding sample complexities.

Theorem 2 shows the sample complexity of CS-LUCB in terms of finding an $\epsilon$-optimal formula for any $\epsilon \geq 0$. In my opinion, it is important to show the lower bound, and it is possible to find such result in the existing BAI literature. Theorem 3 only proves the sample complexity of CS-SE for $\epsilon=0$, and how about the sample complexity for general $\epsilon \geq 0$? I guess it is not hard to extend the current result. In Corollary 2, is it true that we have the first bound only if $\nu_\eta \leq -1/2$ instead of $-1/4$?

I suggest doing a more comprehensive literature review; for example, [1] (and some references therein) needs to be cited and compared in detail.

Here are some minor comments. On line 249, there is a redundant "s" in the word "miss-classified". In the expression of $\nu$ on lines 280 and 282, $\nu_\eta$ should replace $\nu_\theta$.

[1] Masahiro Kato, Takuya Ishihara, Junya Honda, and Yusuke Narita, Adaptive Experimental Design for Efficient Treatment Effect Estimation, https://arxiv.org/abs/2002.05308, 2020.

**Time Spent Reviewing:**

~8 hours

---

> ### Author Response · Authors · 2021-08-12
> **Thanks for the review!**
>
> Thanks for your thoughtful review: your description of our contributions is very accurate.
>
> __Lower Bounds__
>
> We opted for the simplest bandit complexity results, as the aim of this submission was to bring best-arm-identification techniques to the area of selection of identification formulas in the causal inference literature, but we agree that more thorough sample complexity results would improve our contribution. However, extending the sample complexity proof of CS-SE to $\epsilon>0$ is not that straightforward. In the original paper, the authors computed the sample complexity by deriving the number of samples needed to ensure that all the remaining arms had confidence interval width less than $\epsilon$. Because the confidence bounds were deterministic, the number of required samples is deterministic as well. In contrast, we only assume confidence sequences that decay to zero; we would need to find the number of samples required for each arm to have confidence width less than $\epsilon>0$. However, these numbers are random variables (in particular, they are stopping times), so we would need high probability bounds. We chose not to include such a result because of the added complexity.
>
> In general, the flexibility of using confidence sequences makes finding lower bounds difficult: the sample complexities depend on how quickly the widths of the confidence sequence decrease as well as the gaps. For example, the BAI problem with general confidence sequences does not easily allow for implementations of the optimal fixed-confidence BAI algorithms. Track-and-stop style algorithms rely on parametric assumptions about the reward distributions while all we can guarantee is a frequentist confidence sequence on the reward. Minimax-optimal algorithms, such as lil’UCB and Exponential Gap Elimination, cannot be easily transported either for the following reasons. The stopping condition of lil’UCB is triggered when one arm is sampled at least a constant proportion, but the proof of correctness requires that the confidence bound widths shrink at the same rate for every arm, which is not necessarily true in our framework. The same difficulty arises in Exponential Gap Elimination and even Median Elimination; both algorithms also assume a uniform rate of delay of the bound widths across arms. Hence, we were forced to use the simpler algorithms LUCB and SE, which allow us to derive correctness and sample complexity results when the confidence intervals decay at arbitrary speeds. The difficulty of an instance will need to be a function of the shape of the entire confidence sequence, so it will only be easy to adapt the lower bounds to the case where all the confidence sequences are decaying at the same rate.
>
> __Corollary 2__
>
>  The requirement that $\nu_\eta\leq -1/4$ is sufficient for the familiar complexity. $O(\sum_k\Delta_k^{-2})$ is correct; the second-order dependence on the error of the nuisance function estimator is shown by Theorem 1. What isn’t correct is the second part of that displayed equation: it should be $O( \sum_k \Delta_k^{- 1/ 2 \nu_eta})$. Thanks for catching this typo.
>
> __ADR__
>
> Thanks for the literature pointer! Unfortunately, we cannot see how the ADR estimator from Kato et al. 2020 could be used in our setting. Kato et al. 2020 show that asymptotic normality can hold when the interventions are chosen in a data-dependent way. In contrast, our data are observational, as we never change the treatment assignment/intervention and only change which covariates we observe; therefore, our observations remain i.i.d.
> Nevertheless, the work from Kato et al. 2020 opens up a very interesting question: how can we select the best estimator when we may also intervene? More generally, how can we establish finite-sample confidence bound on cross-fitted estimators? Such results would have many applications beyond our setting. We will include a discussion of the adaptive sampling literature.

---

### Decision · Program_Chairs · 2021-09-28

**Decision:**

Accept (Poster)

**Comment:**

The reviewers came to consensus that the techniques materials of this paper is standard and the contribution is somewhat limited, though the introducing the idea of using BAI for causal effect identification is novel. Some reviewers also point out the writing issues and they must be thoroughly addressed to make the paper acceptable for publication.

**Consistency Experiment:**

NeurIPS has a long history of experimentation. In 2014, NeurIPS ran an experiment in which 10% of submissions were reviewed by two independent committees to quantify the randomness in the review process. This year, we repeated a variant of this experiment to see how the quality of the review process has changed over time.  This paper was part of the experiment and was therefore assigned to two committees (consisting of reviewers, an Area Chair, and a Senior Area Chair) that reached independent decisions.  If both committees made the same recommendation, this recommendation was followed. If a single committee recommended acceptance, the paper was accepted (with the exception of a few cases in which the other committee identified what we considered a fatal flaw, e.g., an error in a key result).

This copy’s committee reached the following decision: **Reject**

The other committee assigned to the paper recommended **Accept (Poster)**.  You can find the other set of reviews, along with any follow up discussion with the authors here:
https://openreview.net/forum?id=zqo2sqixxbE